# A new metabolic model of *Drosophila melanogaster* and the integrative analysis of Parkinson's disease

Müberra Fatma Cesur[1], Arianna Basile[2], Kiran Raosaheb Patil[2], Tunahan Çakır[1]

High conservation of the disease-associated genes between flies and humans facilitates the common use of *Drosophila melanogaster* to study metabolic disorders under controlled laboratory conditions. However, metabolic modeling studies are highly limited for this organism. We here report a comprehensively curated genome-scale metabolic network model of *Drosophila* using an orthology-based approach. The gene coverage and metabolic information of the draft model derived from a reference human model were expanded via *Drosophila*-specific KEGG and MetaCyc databases, with several curation steps to avoid metabolic redundancy and stoichiometric inconsistency. Furthermore, we performed literature-based curations to improve gene–reaction associations, subcellular metabolite locations, and various metabolic pathways. The performance of the resulting *Drosophila* model (8,230 reactions, 6,990 metabolites, and 2,388 genes), iDrosophila1 (https://github.com/SysBioGTU/iDrosophila), was assessed using flux balance analysis in comparison with the other currently available fly models leading to superior or comparable results. We also evaluated the transcriptome-based prediction capacity of iDrosophila1, where differential metabolic pathways during Parkinson's disease could be successfully elucidated. Overall, iDrosophila1 is promising to investigate system-level metabolic alterations in response to genetic and environmental perturbations.

## Introduction

*Drosophila melanogaster* is a well-known model organism with highly tractable genetics for gaining insight into human metabolism. It contains the counterparts of various essential human systems such as the central nervous system, gastrointestinal system, kidney (Malpighian tubule in fly), adipose tissue (fly fat body), and liver (fly oenocytes) (Capo et al, 2019). In addition to this conservation, shorter lifespan (transition from embryo to adulthood within 10–14 d), rapid generation time, large numbers of progeny, a well-defined genome, substantially less genetic redundancy, and the availability of advanced genetic tools for this organism have encouraged researchers to investigate many aspects of human diseases via *Drosophila* (Mizuno et al, 2011; Mirzoyan et al, 2019). Highly conserved disease pathways in humans have been extensively analyzed based on the *Drosophila* genes, which share nearly 75–77% homology with disease-associated human genes (Reiter et al, 2001; Pandey & Nichols, 2011; Mirzoyan et al, 2019).

Genome-Scale Metabolic Network (GMN) models facilitate the mathematical representation of biological knowledge about cellular metabolism through the inclusion of all known chemical reactions, metabolites, and genes for an organism. The genes are linked to the associated reactions via conditional statements in Boolean logic, which are known as gene–protein-reaction (GPR) rules (Gu et al, 2019). Metabolic models hold promise to address numerous scientific questions and open novel avenues for identifying potential drug targets, detecting putative biomarkers, in silico metabolic engineering, pan-reactome analyses, understanding metabolic disorders, and modeling host–pathogen interactions (Oberhardt et al, 2013; Nielsen & Keasling, 2016; Gu et al, 2019). Therefore, a large number of GMN models have been developed so far for prokaryotic and eukaryotic model organisms (Gu et al, 2019). Despite impressive experimental efforts and evolving technologies, there is only one recent metabolic model representing comprehensive *D. melanogaster* metabolism. This model called Fruitfly1 (12,056 reactions, 8,132 metabolites, and 2,049 genes) (Wang et al, 2021) was derived from the generic human GMN (Human1) (Robinson et al, 2020) based on gene orthology information from the Alliance of Genome Resources (The Alliance of Genome Resources Consortium, 2020). Another comprehensive *Drosophila* model (BMID000000141998) was developed in 2013 without any manual curations (Büchel et al, 2013). It consists of 6,198 reactions, 2,873 metabolites, and 4,020 genes. Despite the high gene coverage of this model, it allows the synthesis of all biomass components even without any consumption of carbon sources (Büchel et al, 2013; Schönborn et al, 2019). In addition, two curated small-scale metabolic models (flight muscle and larval models) are available for this organism. They represent only the core metabolism of *Drosophila* (Feala et al, 2007; Schönborn et al, 2019). The flight muscle model (194 reactions, 188 metabolites, and 167 genes) was developed in 2007 to elucidate cellular adaptation

[1]Systems Biology and Bioinformatics Program, Department of Bioengineering, Gebze Technical University, Kocaeli, Turkey [2]Medical Research Council Toxicology Unit, University of Cambridge, Cambridge, UK

Correspondence: tcakir@gtu.edu.tr

mechanisms against hypoxia and it was revised in 2008 (Feala et al, 2007; Coquin et al, 2008). More recently, the second metabolic model called FlySilico (363 reactions, 293 metabolites, and 261 genes) was built to simulate larval development (Schönborn et al, 2019). The core models are not suitable for studying the complex metabolism of the fruit fly.

Comprehensive metabolic network models can be developed considering genetic similarities between organisms. This semiautomatic reconstruction approach begins with the creation of a draft model based on a template model. The template model should have a high genetic similarity with the organism of interest. The genes of this reference network are replaced by their orthologous counterparts in the target organism. In this way, existing information about gene associations is transferred to reconstruct new GMN models (Khodaee et al, 2020). To date, orthology-based GMN models have been developed for several organisms because of the high degree of genomic homology between humans and these model organisms (Sigurdsson et al, 2010; Blais et al, 2017; Khodaee et al, 2020; Wang et al, 2021). In the current study, we developed a GMN model of *D. melanogaster* using a curated version of the Human Metabolic Reaction 2 (HMR2) model (Mardinoglu et al, 2014), as the template (Radic Shechter et al, 2021; Zirngibl, 2021). This HMR2-based draft model was reconstructed through the orthology-based mapping of *Drosophila* genes. Then, it was expanded based on metabolic information in KEGG and MetaCyc databases. Manual curation steps were performed for each model component in several steps of the reconstruction process. Thus, we reconstructed a comprehensively curated *D. melanogaster* model called iDrosophila1. This GMN model was analyzed in terms of its phenotypic prediction ability and gene essentiality prediction. Furthermore, iDrosophila1 was shown to represent metabolic changes consistent with Parkinson's disease (PD). Overall, we believe that iDrosophila1 can enable extensive characterization of fly metabolism and the study of the molecular basis of complex human diseases.

## Results and Discussion

Here, we developed a comprehensive GMN model for *D. melanogaster* using available metabolic information. In the reconstruction process, extensive model curations associated with metabolic redundancy, gene/compound name standardization, and missing/incomplete components were performed. These reaction-centric, metabolite-centric, and gene-centric curation steps were commonly applied for both the draft metabolic network and the KEGG-MetaCyc metabolic network as explained in the Materials and Methods section. Thus, we aimed to avoid any inconsistencies and redundancies in the reconstructed networks by revising each metabolic component (reactions, metabolites, and genes). Additional curations were also applied, if necessary. Overall, the model reconstruction process is summarized in Fig 1.

### Draft reconstruction of *Drosophila* metabolic networks

As the starting point, we reconstructed a draft *D. melanogaster* model based on a recent template human model using an orthology-based approach. The template model, an improved version of the HMR2, incorporates several improvements based on the experimental results and the available sources (e.g., previous GMN models) for accurate phenotype predictions and appropriate contextualization (Radic Shechter et al, 2021; Zirngibl, 2021). The HMR2 is the updated version of the human metabolic reaction (HMR) database (Agren et al, 2012), derived from the Edinburgh human metabolic network (Hao et al, 2010), Recon1 (Duarte et al, 2007), and external databases (Mardinoglu et al, 2014). The modifications in the HMR2-derived template human model mainly include the addition of mitochondrial intramembrane space, the removal of atomically imbalanced reactions, the curation of GPR associations, and the revision of reactions from the beta-oxidation pathway. The addition of the mitochondrial intramembrane space is particularly promising for predictions on respiratory ATP synthesis (Radic Shechter et al, 2021; Zirngibl, 2021). We further curated the GPR rules of the template model based on the information about protein complexes in the iHsa model (Blais et al, 2017). Thus, over 300 GPR associations were curated. In the reconstruction process of the draft fly model, we used metabolic information in the curated human model. In this process, the GPR rules of the template model were converted to *Drosophila* GPR rules via orthology-based gene mapping at a high confidence level, as explained in the Materials and Methods section. At least one *Drosophila* ortholog was identified for 1,986 out of 2,479 human model genes (Table S1). Only one *Drosophila* ortholog was assigned to ~90% of the human genes, whereas the remaining genes were matched with multiple orthologs. The presence of the same *Drosophila* orthologs for some human genes led to redundancy in the GPR rules in the draft network. These genes were revised to remove redundancy in the model. Gene-associated human model reactions with no *Drosophila* orthologs were not included in the draft model. We also kept the nonenzymatic reactions in the model. This approach enabled the generation of an HMR2-based draft *Drosophila* model through the transfer of information about gene associations. Importantly, the HMR2-based template human model contains eight intracellular compartments (cytosol, nucleus, Golgi apparatus, endoplasmic reticulum, mitochondria, mitochondrial intermembrane space, lysosome, and peroxisome) along with their Gene Ontology IDs, and this information was also transferred to the draft *Drosophila* model. The reconstructed draft model includes 6,873 reactions, 4,856 metabolites, and 1,321 genes.

Using the metabolic information in KEGG and MetaCyc databases, a KEGG–MetaCyc metabolic network was also generated for *D. melanogaster*. We encountered two major issues with this network including (1) a need for the modification of metabolite names and (2) the lack of compartmentalization. As highlighted before, the main contribution of the KEGG–MetaCyc network is to expand gene coverage and available metabolic information in the HMR2-based draft model. Therefore, metabolite names in the KEGG–MetaCyc network should be compatible with the draft model for proper merging. To do so, we updated compound names in the KEGG–MetaCyc network by replacing them with the counterparts in the HMR2-based draft model (Fig S1). This step is critical to reduce potential metabolic redundancy in the merged model. To avoid metabolic redundancy, we also removed the KEGG–MetaCyc genes

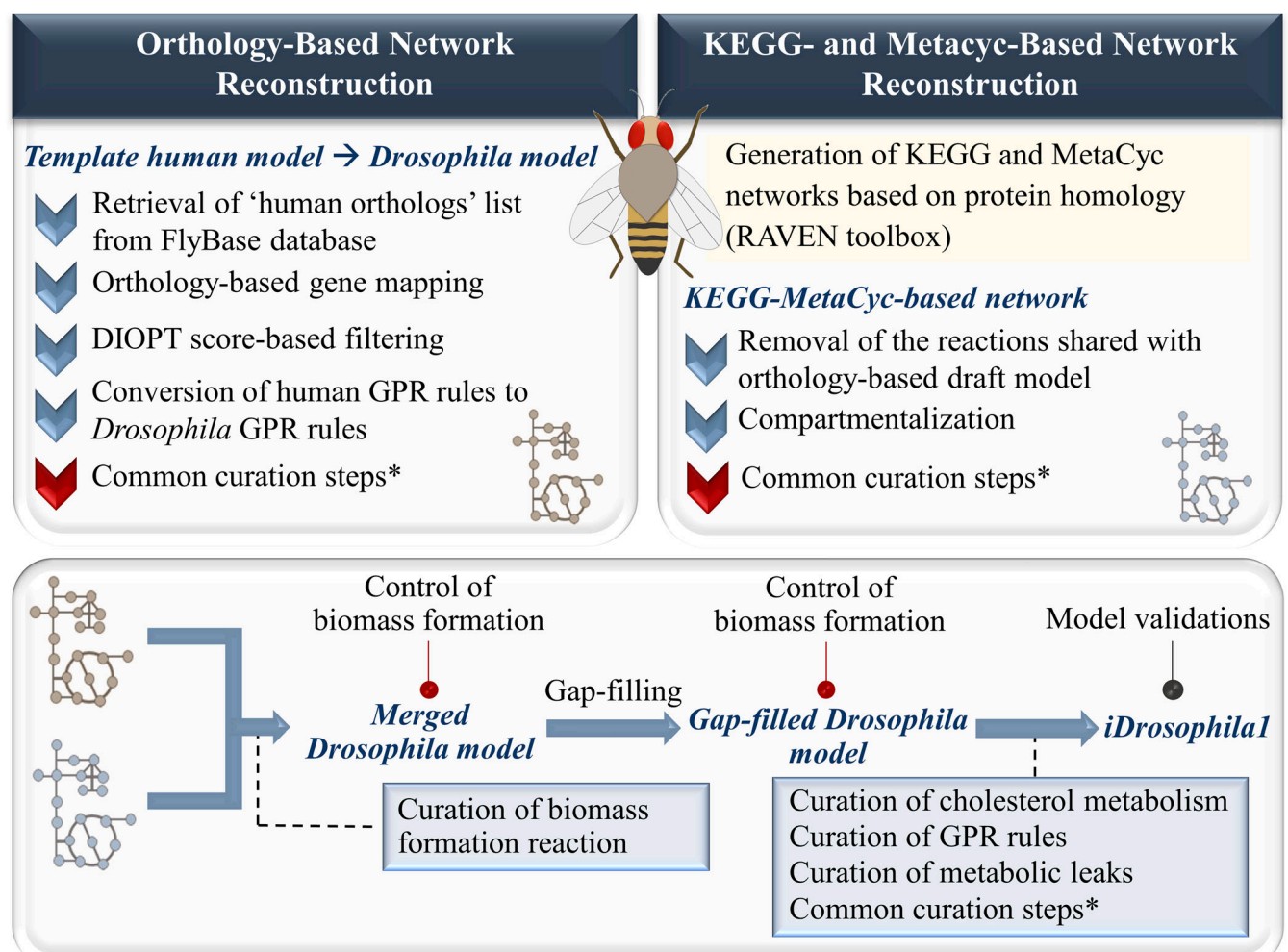

**Figure 1. The summarized reconstruction process of the genome-scale metabolic network model iDrosophila1 for *Drosophila melanogaster*.**
Commonly applied reaction-centric, metabolite-centric, and gene-centric curations are detailed in the Materials and Methods section.

that were shared by the draft model, leading to a KEGG–MetaCyc-specific metabolic network. In this step, we determined 1,197 KEGG–MetaCyc-specific genes and 811 common genes. The KEGG–MetaCyc-specific genes were characterized by identifying enriched KEGG pathways. In addition to the fundamental pathways (e.g., carbohydrate, amino acid, fatty acid, and cofactor metabolism), drug and xenobiotic metabolic processes were found among the enriched pathways (Table S2). Xenobiotics are any exogenous life-threatening, toxic compounds (e.g., pharmaceuticals, pesticides, and pollutants) to which organisms are exposed (Misra et al, 2011; Trinder et al, 2017). The role of xenobiotics in neurodegenerative disorders like Alzheimer's disease (AD) and PD was reported (Chin-Chan et al, 2015; Bjørklund et al, 2020). Accordingly, rotenone and paraquat are commonly used to induce a PD-like phenotype (e.g., movement disorders and loss of dopaminergic neurons) in *Drosophila* by triggering oxidative stress (Coulom & Birman, 2004; Hosamani and Muralidhara, 2010; Muñoz-Soriano & Paricio, 2011; Nagoshi, 2018). Thus, xenobiotics are important to model PD phenotype in flies. As a result, the reconstruction of the KEGG–

MetaCyc network supported extended metabolic information about *D. melanogaster*. This may be promising in future analyses to shed light on the molecular mechanisms underlying a variety of human diseases.

Considering the Gene Ontology IDs in the draft *Drosophila* model, we compartmentalized the KEGG–MetaCyc-specific metabolic network because it did not include compartment information. To do so, we first identified the compartments of *Drosophila* gene products in the KEGG–MetaCyc network via available resources (COMPARTMENTS, FlyBase, GLAD, QuickGO, AmiGO 2, UniProt, Reactome, and a mass spectrometry-based study). The subcellular localization information was mapped to each gene product in the network. Missing compartments were predicted by the CELLO2GO web server for the cutoff of $10^{-5}$ and so we generated a gene–compartment pair list. Using this approach, multiple compartments were assigned to many gene products (Table S3). This compartment dictionary allowed the assignment of subcellular localization(s) to each reaction based on the GPR associations. In this step, if a gene product has multiple compartments, we assumed that the related

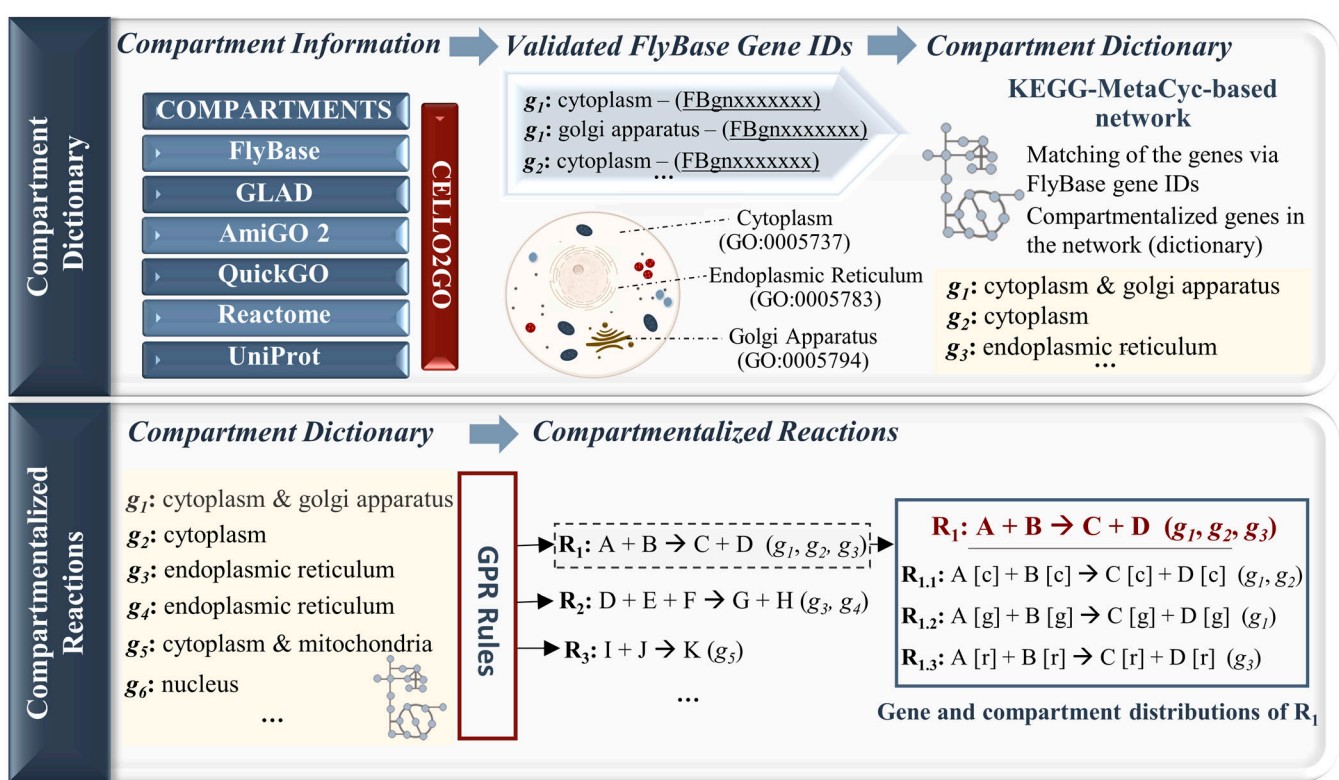

**Figure 2. A general framework applied for the compartmentalization of the KEGG–MetaCyc network.**
This process starts with the retrieval of compartment information along with validated FlyBase gene IDs from different sources. The compartment information is mapped to the genes in the KEGG–MetaCyc network to generate a compartment dictionary. Based on the GPR associations, the compartment information is transferred from the genes to the reactions. In the given toy network, the reaction $R_1$ provides the conversion of metabolites A and B to C and D, and it is catalyzed via the enzymes encoded by three different genes: $g_1$, $g_2$, and $g_3$. In the compartmentalization process, $R_1$ is added to the network in a repeated manner for each gene compartment (c: cytosol, g: Golgi apparatus, and r: endoplasmic reticulum). The genes are subsequently distributed to the reactions ($R_{1.1}$, $R_{1.2}$, and $R_{1.3}$) derived from $R_1$ based on their compartment information.

reaction should be repeated in the model for each compartment. Thus, new reactions and metabolites were added to the network according to the subcellular locations of the corresponding gene product, if necessary. In addition, if the genes catalyzing a reaction have different compartments, multiple compartments were assigned to this reaction and the corresponding genes were distributed based on their localizations. A general framework of the network compartmentalization process is illustrated in Fig 2 for the sake of clarity. The compartmentalized KEGG–MetaCyc-specific network consists of 1,077 genes and 3,511 metabolites involved in 2,015 enzymatic reactions.

### Combining draft *Drosophila* metabolic networks and additional curations

The HMR2-based draft *Drosophila* model was merged with the compartmentalized KEGG-MetaCyc network. Thus, the number of genes in the draft model was elevated to 2,398. The ability of the merged model to produce biomass was assessed. We determined that many cofactors and vitamins could not be produced. A gap-filling algorithm was therefore applied to ensure the production of all biomass precursors. It allowed the addition of 29 new reactions (Table S4) to the merged model. The newly added reactions

enabled filling in the gaps related to cofactor and vitamin metabolism. In the next step, the gap-filled model was curated in terms of cholesterol and apocytochrome-C metabolism, GPR rules, metabolic leaks, and the commonly applied curation steps described in the Materials and Methods section.

Cholesterol acts as the major structural component of the *Drosophila* membrane and the precursor of steroid hormones. Steroid production is crucial in the regulation of developmental processes, which are required to generate an adult organism (Niwa & Niwa, 2011; Danielsen et al, 2016) (Fig 3A). On the other hand, *D. melanogaster* has proved to be a cholesterol auxotroph, which means the inability for de novo cholesterol synthesis because of an incomplete cholesterol biosynthesis pathway (Vinci et al, 2008; Knittelfelder et al, 2020). In mammals, 3-hydroxy-3-methylglutaryl CoA reductase enzyme converts 3-hydroxy-3-methylglutaryl CoA into mevalonate. Using a set of enzymes, this compound is converted to farnesyl pyrophosphate. Santos & Lehmann (2004) uncovered several fly orthologs catalyzing this pathway, which is branched to the isoprenoid synthesis process (Fig 3B). The farnesyl pyrophosphate can also be directed to the cholesterol synthesis branch in mammals. Most human genes involved in this branch (from farnesyl pyrophosphate to cholesterol) are not conserved in flies (Santos &

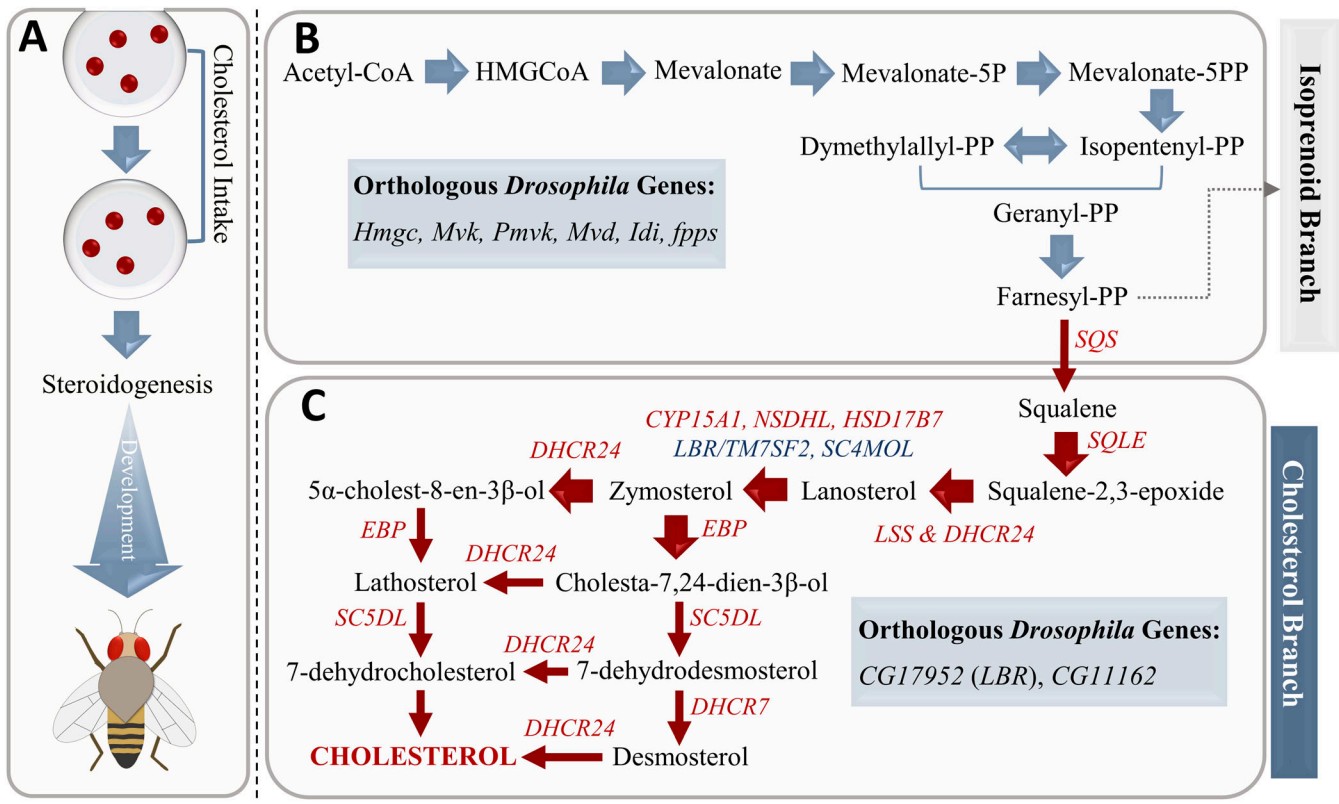

**Figure 3. Cholesterol metabolism in *Drosophila melanogaster*.**
**(A)** *Drosophila* needs cholesterol as a precursor to produce steroid hormones, which are crucial for the regulation of developmental processes. The initial steps of the cholesterol metabolic process including farnesyl-PP synthesis are conserved in all animals. **(B, C)** The farnesyl-PP can be metabolized via two main pathways: (B) the isoprenoid branch and (C) the cholesterol branch in mammals. However, *Drosophila* does not include many genes in the cholesterol branch (indicated in red) and conserved genes in the cholesterol metabolism are indicated in blue. These metabolic gaps underlie cholesterol auxotrophy in the fruit fly.

Lehmann, 2004; Zhang et al, 2019) (Fig 3C). Hence, *Drosophila* must acquire sterols from dietary components (Vinci et al, 2008; Danielsen et al, 2016; Knittelfelder et al, 2020). Zhang et al (2019) identified only two cholesterol synthesis genes in humans (*SC4MOL* and *LBR/TM7SF2*) with *Drosophila* orthologs, whereas there are many non-orthologous human genes (*SQS*, *SQLE*, *LSS*, *CYP51A1*, *NSDHL*, *ERG27*, *DHCR24*, *EBP*, *SC5DL*, and *DHCR7*) (Zhang et al, 2019) (Fig 3C).

We determined the cholesterol biosynthesis reactions in the draft *Drosophila* model that correspond to the inactive cholesterol branch. For these reactions, we first identified the corresponding *Drosophila* genes that were incorrectly defined as human orthologs. Accordingly, two *Drosophila* genes were found to be assigned as the orthologs of *NSDHL* gene-encoding sterol-4-alpha-carboxylate 3-dehydrogenase (decarboxylating) and *DHCR7* gene-encoding 7-dehydrocholesterol reductase in human. Of these *Drosophila* genes, *CG7724* (FBgn0036698), associated with steroid biosynthesis, was matched with *NSDHL* (ENSG00000147383) at the maximum DIOPT score of 3. Because *Drosophila* was reported to lack *NSDHL* ortholog (Zhang et al, 2019), we removed the corresponding reactions from the *Drosophila* model. These reactions are responsible for the synthesis of 3-keto-4-methylzymosterol, 5α-cholesta-8,24-dien-3-one, 4α-methyl-5α-cholesta-8-en-3-one, and

5α-cholesta-8-en-3-one compounds (Table 1). Another incorrectly assigned *Drosophila* gene, *LBR* (FBgn0034657)-encoding lamin B receptor, was matched with *DHCR7* (ENSG00000172893) at the maximum DIOPT score of 5. Because *DHCR7* was also defined as a non-ortholog, we excluded the related reactions (HMR_1519 and HMR_1565) from the model (Table 1). The reaction "HMR_1519" is responsible for the formation of desmosterol from 7-dehydrodesmosterol and the reaction "HMR_1565" allows the conversion of provitamin D3 (7-dehydrocholesterol) to cholesterol. Furthermore, eight non-conserved cholesterol biosynthesis reactions that were included in the *Drosophila* model in the gap-filling step were subsequently removed from the model. These reactions are also listed in Table 1. In conclusion, we curated the cholesterol metabolism in the *Drosophila* model through the removal of the non-conserved human reactions. This step was necessary to mimic the cholesterol auxotrophy of flies. In addition to the cholesterol metabolism, we revised apocytochrome-C metabolism. To this end, the missing metabolite, apocytochrome-C, was added to the reaction "HMR_4762" related to porphyrin metabolism. Four reactions involved in the apocytochrome-C metabolism were also added to the *Drosophila* model from HMR2.

In the next step, we curated the GPR rules lacking protein complex information. This was achieved based on the *Drosophila*

**Table 1.  Metabolic network-driven investigation of non-conserved cholesterol biosynthesis reactions between humans and flies.**

| Cholesterol biosynthesis gene | Incorrectly matched *Drosophila* gene | Corresponding *Drosophila* model reaction |
|---|---|---|
| *SQLE* (ENSG00000104549) | NA (gap–filling-aided addition) | HMR_1470: squalene 2,3-oxide (squalene 2,3-epoxide) synthesis from squalene |
| *LSS* (ENSG00000160285) | NA (gap–filling-aided addition) | HMR_1473: conversion of squalene 2,3-oxide (squalene 2,3-epoxide) to lanosterol |
| *CYP51A1* (ENSG00000001630) | NA (gap–filling-aided addition) | HMR_1477: 4,4-dimethyl-14$\alpha$-hydroxymethyl-5$\alpha$-cholesta-8,24-dien-3$\beta$-ol synthesis from lanosterol |
| | | HMR_1478: 4,4-dimethyl-14$\alpha$-formyl-5$\alpha$-cholesta-8,24-dien-3$\beta$-ol synthesis |
| | | HMR_1479: 4,4-dimethyl-5$\alpha$-cholesta-8,14,24-trien-3$\beta$-ol synthesis |
| *NSDHL* (ENSG00000147383) | *CG7724* (FBgn0036698) DIOPT score: 3 orthology-aided addition | HMR_1495: 3-keto-4-methylzymosterol synthesis |
| | | HMR_1496: 3-keto-4-methylzymosterol synthesis |
| | | HMR_1505: 5$\alpha$-cholesta-8,24-dien-3-one synthesis |
| | | HMR_1546: 4$\alpha$-methyl-5$\alpha$-cholesta-8-en-3-one synthesis |
| | | HMR_1551: 5$\alpha$-cholesta-8-en-3-one synthesis |
| *DHCR24* (ENSG00000116133) | NA (gap–filling-aided addition) | HMR_1570: cholestenol synthesis from zymosterol |
| EBP (ENSG00000147155) | NA (gap–filling-aided addition) | HMR_1553: conversion of cholestenol to lathosterol |
| *SC5DL* (ENSG00000109929) | NA (gap–filling-aided addition) | HMR_1557: provitamin D3 (7-dehydrocholesterol) synthesis from lathosterol |
| *DHCR7* (ENSG00000172893) | *LBR* (FBgn0034657) DIOPT score: 5 orthology-aided addition | HMR_1519: desmosterol synthesis from 7-dehydrodesmosterol |
| | | HMR_1565: cholesterol synthesis from provitamin D3 (7-dehydrocholesterol) |

The non-conserved reactions that were incorrectly included in the draft Drosophila model via gap-filling or orthology-aided approach were determined based on the non-orthologous human genes in the literature. These reactions were excluded from the draft Drosophila model in the next step.

network including 556 protein complexes, which was developed by Guruharsha and colleagues (Guruharsha et al, 2011). We linked the fly genes with "AND" operators if they were found in the same protein complex. Based on the FlyBase gene group list and previous studies (Van den Berghe et al, 1997; Santos & Lehmann, 2004; Allan et al, 2005; Wahl et al, 2005; Avval & Holmgren, 2009; Grant et al, 2010; Pavlovic & Bakovic, 2013; Tang & Zhou, 2013; Kemppainen et al, 2014; Attrill et al, 2016; Kovacs et al, 2018; Marygold et al, 2020b; Rhooms et al, 2020), we curated additional GPR rules including energy metabolism-related gene associations such as mitochondrial complexes I–V. The FlyBase gene groups consist of manually curated members within distinct gene families, the subunits of protein complexes, and other functional gene sets (Attrill et al, 2016; Marygold et al, 2016). We linked the genes encoding complex subunits (except for the paralogous genes) with "AND" operators. This further refined the protein complex information in the GPR associations and facilitated the addition of the missing genes in the complexes. Modifications in the mitochondrial complexes are particularly crucial for an improved representation of energy metabolism in condition-specific metabolic networks. Because any impairments in mitochondrial function may lead to disrupted cellular phenomena (e.g., defective energy metabolism, elevated reactive oxygen species levels, and altered apoptotic signals), mitochondrial dysfunction was reported among the well-known causes of many neurodegenerative disorders (Wu et al, 2019;

Monzio Compagnoni et al, 2020; Rhooms et al, 2020). Therefore, curation of the related GPR associations is important for a more accurate characterization of such diseases. For accurate model simulations, we also updated biomass formation reaction based on the FlySilico model (Schönborn et al, 2019) and the literature.

The updated GMN model was further revised through the reaction-centric, metabolite-centric, and gene-centric curation steps (see the Materials and Methods section). One significant modification is the removal of duplicated reactions. The duplicated reactions were identified via reaction comparisons by ignoring H$^+$, H$_2$O, and free inorganic phosphate (P$_i$). Given that a relatively flexible compartmentation procedure was applied in the KEGG–MetaCyc network, compartment information was excluded for the comparisons between KEGG–MetaCyc-derived reactions and template human model-derived reactions in the draft *Drosophila* model. In this way, we eliminated incorrectly compartmentalized KEGG–MetaCyc reactions from the *Drosophila* model by carefully examining each reaction match. This also enabled the curation of the leaking ATP problem in the model by removing the incorrectly compartmentalized uridine 5′-monophosphate phosphohydrolase, succinate dehydrogenase (ubiquinone), and NADH-dehydrogenase reactions. We searched for the presence of other leaking energy metabolites in the draft model and identified four additional leaking metabolites including NADH, NADPH, FADH$_2$, and H$^+$. Double-reaction deletion knockouts were performed using the flux

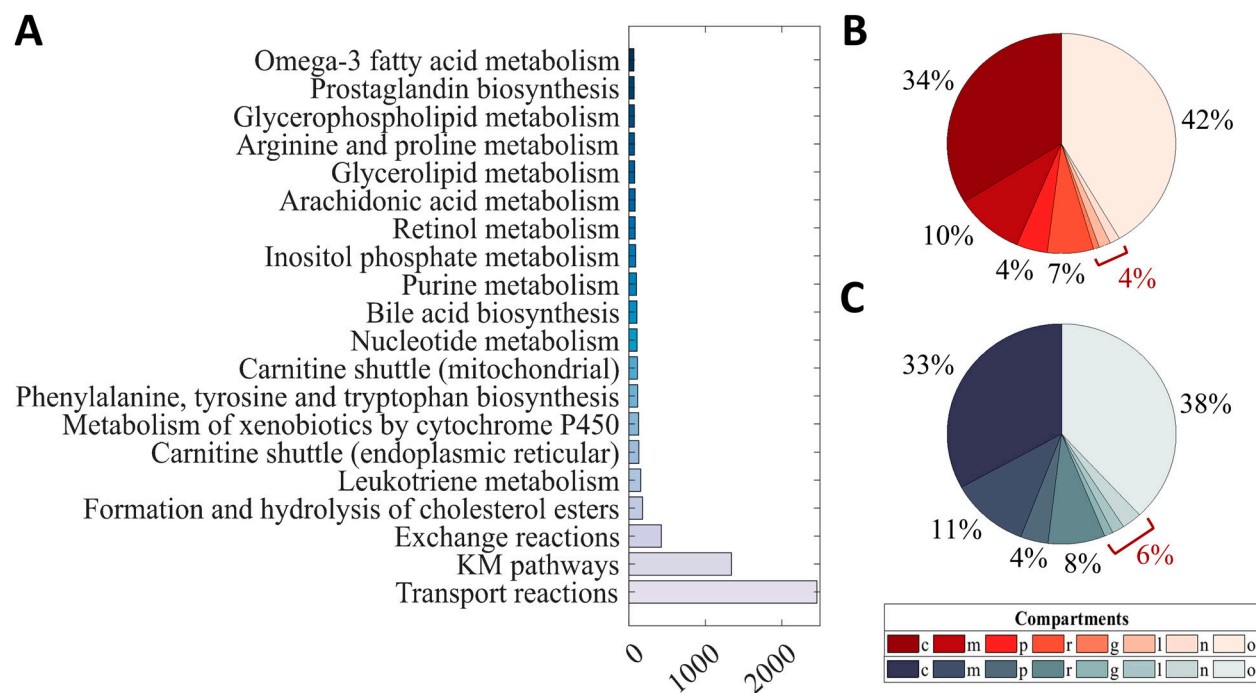

**Figure 4. Distribution of the pathway and compartment information in the iDrosophila1 model.**
**(A)** HMR2-based metabolic pathways in the model are ordered according to their frequency and the top 10 pathways are represented. **(B, C)** The pie chart indicates the percentage of compartments in the (B) template human model and (C) iDrosophila1. The sum frequency values (given in red) are shown for the compartments with low frequencies. (Abbreviations: c, cytosol; m, mitochondria; p, peroxisome; r, endoplasmic reticulum; g, Golgi apparatus; l, lysosome; n, nucleus; o, others [exchange and transport reactions]).

balance analysis (FBA) to identify the reactions associated with metabolic leaks. These reactions were manually examined and they were classified as incorrectly compartmentalized reactions or redundant reactions. The reason for this metabolic redundancy was the undetected duplicated reactions derived from the KEGG–MetaCyc model and the template human model because of small differences in metabolite names. Therefore, we removed one of each duplicated reaction to prevent metabolite leakage. The final model, called iDrosophila1, includes 8,230 reactions (5,787 enzymatic and 2,443 nonenzymatic), 6,990 metabolites, and 2,388 genes. The compatibility of the model with recommended standards was assessed using the MEMOTE test suite. Norsigian et al reported the MEMOTE scores of 108 reconstruction models in the BiGG Models database, most of which are prokaryotic models (Norsigian et al, 2020). The iDrosophila1 model has a comparable score with the other 108 models (Fig S2).

The iDrosophila1 reactions with missing subsystem (pathway) information were investigated through the KEGG database. The pathway information of any reactions which could be accessed was included in the model. The pathways derived from the KEGG–MetaCyc network were denoted as "KM pathways." Analysis of the pathway information in the model pointed to the prevalence of major biological processes (e.g., lipid, amino acid, and nucleotide metabolisms) and cholesterol ester metabolism (Fig 4A). Especially, lipid metabolic pathways were found to have high frequencies. Cholesterol ester and triacylglycerol are the storage lipids accumulated in *Drosophila* fat body cells (Liu & Huang, 2013). These dietary lipids are converted into free fatty acids, sterols, and

monoacylglycerols that are absorbed by the cells of the fly intestine under normal feeding conditions. The resynthesized triacylglycerols are then packaged into lipoproteins together with carrier proteins, cholesterols, and cholesterol esters for transportation along the body. In this way, they can be used or stored by the tissues including adipose and liver. The presence of excess lipids induces the utilization of cholesterol esters and triacylglycerols as energy-supplying fuels (Sieber & Thummel, 2012). Xenobiotic metabolism was determined to be another prominent pathway in the iDrosophila1 model. As highlighted before, xenobiotics are natural or synthetic life-threatening compounds that must be handled by animal cells via sequestration or metabolic degradation. Modification of the xenobiotics by phase I enzymes (e.g., cytochrome P450 monooxygenase and esterases) occurs in the first step of their metabolic detoxification (Misra et al, 2011; Trinder et al, 2017; Amichot & Tarès, 2021). In addition to the pathway information, we examined the compartment distribution in the model. We observed a similar compartment distribution profile between the template human model (Fig 4B) and the iDrosophila1 (Fig 4C). Cytosolic reactions and transport reactions were found to have high frequencies in both models, and they were followed by a mitochondrial distribution (Fig 4B and C).

## Prediction of growth phenotypes under increasing cholesterol and amino acid levels

In nature, *Drosophila* feeds on fermenting fruits containing high amounts of ethanol and organic acids. On the other hand, a simple

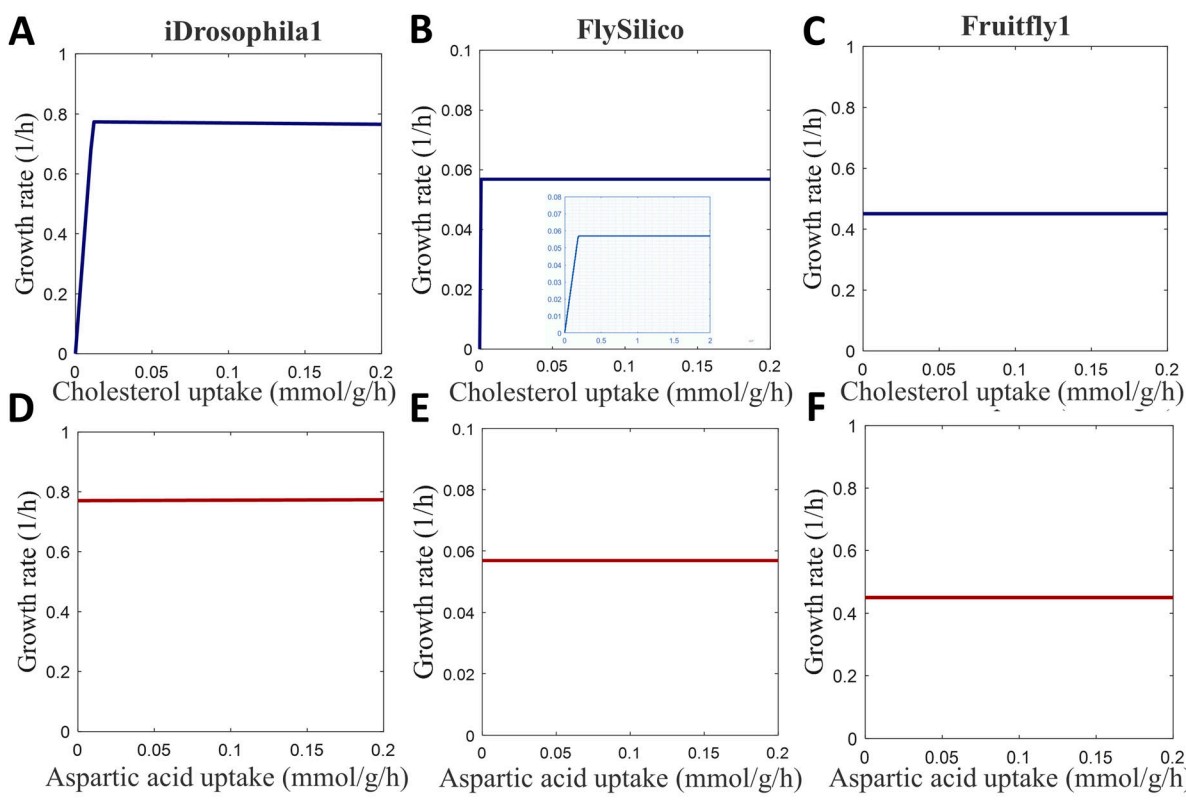

**Figure 5. Growth profiles of *Drosophila* under expanded HD condition supplied with elevating cholesterol and aspartate levels.**
**(A, B, C)** iDrosophila1, (B) FlySilico, and (C) Fruitfly1 models are used to characterize cholesterol-dependent changes in the proliferation. **(B)** The zoom-out view associated with FlySilico simulations demonstrates a smaller uptake range of cholesterol to clearly represent the relationship between exogenous cholesterol supplementation and growth profile. **(D, E, F)** display the effect of aspartate levels on the growth rates in the simulations through iDrosophila1, FlySilico, and Fruitfly1 models, respectively.

diet of sucrose, lyophilized yeast, and weak organic acids was reported as sufficient to raise this organism in laboratory conditions. To mimic this dietary restriction, Piper and colleagues established the holidic diet (HD) with essential and nonessential amino acids, vitamins, cholesterol, sucrose, several metal ions, and lipid precursors. This diet composition was reported to be sufficient for the development of fruit flies (Piper et al, 2014). Schönborn and colleagues simulated larval fly growth on the HD with a yeast-like amino acid ingredient. The researchers reported the predicted larval growth rate as 0.088 h$^{-1}$ via the FlySilico model by employing a set of constraints for the uptake rates (Schönborn et al, 2019). Using the iDrosophila1 model, we also analyzed the growth of *D. melanogaster* with the same uptake constraints (Table S5). For compatibility with the FlySilico model, we fixed the flux of the non-growth-associated maintenance reaction to the estimated value (8.55 mmol ATP g$^{-1}$ h$^{-1}$) (Schönborn et al, 2019). FBA simulation was then performed with the growth maximization in the expanded HD condition. The growth rate was predicted to be 0.040 h$^{-1}$ (Schönborn et al, 2019).

As aforementioned, cholesterol has crucial roles in membrane structure and signaling processes, but flies are unable to synthesize this essential compound (Vinci et al, 2008). To properly mimic the cholesterol auxotrophy of *Drosophila*, we introduced several modifications in iDrosophila1 (Table 1) relying on the literature

(Santos & Lehmann, 2004; Zhang et al, 2019). The capability of the model to accurately predict this phenotypic property was subsequently assessed. In this regard, we allowed flexible intake of the expanded HD compounds by limiting the maximum sucrose uptake rate to ~2.212 mmol g$^{-1}$h$^{-1}$ and supplying the other metabolites at a certain ratio (see the Materials and Methods section). In this condition, the optimal growth rate was predicted as ~ 0.774 h$^{-1}$. We subsequently analyzed the growth profile across varying cholesterol levels from zero to ~1/10 of the sucrose uptake rate. As expected, we did not observe biomass formation for the cholesterol deficiency in the diet. Increasing level of cholesterol positively induced biomass formation until reaching the optimal growth rate (Fig 5A). We also performed this simulation through Fruitfly1 and FlySilico models by applying the same constraints to limit the consumption of the HD compounds. The Fruitfly1 model failed to grow under the expanded HD condition. Therefore, three additional substances (lipoic acid, linoleate, and linolenate) were supplied to allow biomass formation. In addition, the right-hand side of the biomass formation reaction in Fruitfly1 (version 1.1.0) is missing ADP that is required to establish a balance between ATP and ADP. Therefore, we added this compound to the biomass formation reaction of Fruitfly1 for all simulations covered in this study. Note that biomass was not produced in the Fruitfly1 model using measured uptake flux boundaries (Schönborn et al, 2019) (Table S5).

On the other hand, the use of flexible boundary constraints triggered a growth rate of 0.450 h$^{-1}$. FlySilico simulation, on the other hand, resulted in the maximum growth rate of 0.057 h$^{-1}$ for the flexible uptake rates. In the next step, we examined the growth profiles of these models across the increasing cholesterol levels. Elevating cholesterol intake triggered an enhanced growth rate in FlySilico (Fig 5B) in agreement with the iDrosophila1 simulation, whereas this phenotypic feature could not be observed by Fruitfly1 (Fig 5C).

Aspartate is a nonessential amino acid whose deficiency was reported to have no detrimental effect on the lifespan of *Drosophila* (Piper et al, 2014). Schönborn and colleagues reported that the increasing level of the aspartate amino acid did not affect biomass production (Schönborn et al, 2019). We also examined the impact of varying aspartate levels on fly growth in the expanded HD condition in the iDrosophila1, FlySilico, and Fruitfly1 models. The iDrosophila1 model revealed that biomass production was not affected by aspartate depletion thanks to its inherent aspartate biosynthesis system. Besides, supplementation with additional aspartate did not enhance the growth at a considerable level (Fig 5D). This is consistent with FlySilico (Fig 5E) and Fruitfly1 (Fig 5F) simulations (Fig 5E). Furthermore, *Drosophila* features 10 essential amino acids, which were reported to be commonly essential between mammals and insects, except for arginine (Croset et al, 2016; Manière et al, 2020). We affirmed their essentiality using the iDrosophila1 model. The elevating intake of each essential amino acid contributed to an increase in the growth rate (Fig S3). Altogether, we confirmed the growth profile of fly across the varying levels of cholesterol and amino acids using iDrosophila1.

## Prediction of essential *Drosophila* genes

Gene essentiality refers to the indispensability of genes for survival under specific growth conditions. This concept is especially suitable to analyze cell type-specific gene essentiality because of cellular variations. On the other hand, it is often used to evaluate the predictive capabilities of the reconstructed generic metabolic models because of the lack of comprehensive cell-specific information for many eukaryotic organisms. We assessed the prediction performance of the generic iDrosophila1 model by in silico single-gene knockouts, and tissue-specific analyses may result in a higher number of the essential genes. For the gene knockout simulations, we blocked the corresponding reactions for the deletion of each gene. Essential genes were determined considering predicted growth rates. Accordingly, the genes whose deletions led to a significant reduction in growth rate were accepted as essential. For the iDrosophila1 model, we elucidated essential and nonessential gene sets through the FBA approach for unlimited intake of the expanded HD components, leading to 128 essential genes (Table S6A). We performed GO and pathway enrichment analyses to provide an insight into these genes in terms of corresponding biological processes (Table S6B) and pathways (Table S6C). Unsurprisingly, these genes were found to be predominantly associated with biosynthetic processes of crucial cellular substances such as nucleotides, aminoacyl-tRNAs, amino acids, cofactors, and lipids. Enriched KEGG pathways were also identified to be consistent with these biological processes.

We further evaluated the performance of the iDrosophila1 model through the comparison of the essentiality predictions with those obtained by other curated generic fly models. In this process, we introduced two main modifications to the Fruitfly1 model before gene essentiality analysis. The first modification is related to RNA metabolism. Two diverse cytosolic RNA synthesis reactions are present in both iDrosophila1 (HMR_7161 and HMR_7162) and Fruitfly1 (MAR07161 and MAR07162) models. They use nucleoside triphosphates (NTPs) and nucleoside diphosphates (NDPs) as substrates, respectively. Because both molecules contain phosphoanhydride bonds, they are significant energy sources to drive biochemical reactions. NTPs (ATP, GTP, UTP, and CTP) also serve as substrates for nucleic acid biosynthesis by DNA-directed RNA polymerase (RNAP) enzymes (Gottesman & Mustaev, 2019). Each RNAP enables the synthesis of distinct RNA classes from ribosomal RNAs to noncoding RNAs (Marygold et al, 2020a). The Fruitfly1 and iDrosophila1 models have nuclear and mitochondrial genes encoding RNAP subunits. Another reaction associated with RNA metabolism is catalyzed by polynucleotide phosphorylase (PNPase) in the presence of NDPs (Gottesman & Mustaev, 2019; Pajak et al, 2019). These conserved enzymes are responsible for RNA turnover primarily by degrading mitochondrial RNAs (Das et al, 2011; Pajak et al, 2019). Recently, ATP-dependent RNA helicase SUV3 and PNPase enzymes were proposed to form a minimal mitochondrial RNA degradosome complex for mRNA decay in *Drosophila* (Pajak et al, 2019). PNPases (EC 2.7.7.8) were demonstrated to participate in RNA polymerization (non–template-encoded RNA synthesis) leading to the yield of a high P$_i$ level, which can also allow RNA phosphorolysis (Gasteiger et al, 2003; Gottesman & Mustaev, 2019). On the contrary, NTP polymerization by RNAPs occurs irreversibly. In the Fruitfly1 model, RNAP genes were found in the GPR rules of both NDP polymerization (MAR07162) and NTP polymerization (MAR07161) reactions, whereas only PNPase-encoding *Drosophila* gene (FBgn0039846/CG11337) is responsible for the catalysis of NDP polymerization (HMR_7162) in the iDrosophila1 model. To support this, Pajak and colleagues reported that the "CG11337" gene is the fly ortholog of human PNPase (Pajak et al, 2019). The PNPase-mediated NDP polymerization is also consistent with our template human model (Radic Shechter et al, 2021; Zirngibl, 2021) and the iHsa model (Blais et al, 2017). Taken together, we updated the GPR rule of the Fruitfly1 reaction (MAR07162) before in silico gene deletions by replacing the RNAPs with the PNPase. In addition, we modified the reversibility of this reaction to represent its reversible characteristics. The reaction reversibility was also curated in the iDrosophila1 model.

The second modification is related to the GPR association of the nucleo-cytoplasmic DNA transport reaction in Fruitfly1 (MAR08639), where nuclear pore complex genes were assigned. Nuclear pore complexes comprise several copies of ~30 different proteins known as nucleoporins that mediate macromolecular trafficking (e.g., the transport of RNAs, proteins, and ribosomal subunits) between nucleus and cytoplasm and the free diffusion of water, sugars, and ions (Wente & Rout, 2010; Ibarra & Hetzer, 2015; Kuhn & Capelson, 2018). Despite the presence of multiple *Drosophila* nucleoporins (e.g., Nup98 and Nup62) showing dynamic chromatin binding behavior in the nucleoplasm, they only mediate transcriptional regulations (Kuhn & Capelson, 2018). The artificial DNA transport

**Table 2. Comparison of the predictive metrics calculated for the generic fly models to assess their performances in terms of gene essentiality prediction.**

| Metrics | Fly metabolic network models | | |
| --- | --- | --- | --- |
| | iDrosophila1 (This study) | FruitFly1 Wang et al (2021) | FlySilico Schönborn et al (2019) |
| Sensitivity | 0.16 | 0.09 | 0.03 |
| Specificity | 0.96 | 0.96 | 0.97 |
| Precision | 0.74 | 0.72 | 0.50 |
| Accuracy | 0.61 | 0.52 | 0.52 |
| F1 score | 0.26 | 0.16 | 0.05 |
| MCC | 0.20 | 0.11 | 0.01 |

reactions in the iDrosophila1 and Fruitfly1 models reflect the contribution of DNA to cytosolic biomass formation. Hence, we removed the nucleoporin genes assigned to the artificial nucleo-cytoplasmic DNA transport reaction in Fruitfly1. Similar to the template human model (Radic Shechter et al, 2021; Zirngibl, 2021) and the iHsa model (Blais et al, 2017), the iDrosophila1 model does not contain any genes dedicated to the corresponding transport reaction (HMR_8639).

Based on the updated GPR rules, we also determined essential (Table S7A and B) and nonessential gene sets in the expanded HD medium for the Fruitfly1 and FlySilico models. Of the predicted 128 essential genes, iDrosophila1-specific results (n = 64) were pre-dominantly found to be related to the metabolism of nucleotides, cofactors, lipids, amino acids, and aminoacyl-tRNAs. For Fruitfly1-specific essential genes (n = 33), metabolic processes associated with vitamins, cofactors, amino acids, aminoacyl-tRNAs, and nu-cleotide sugars were shown to be significantly enriched. On the other hand, the FlySilico model predicted only six essential genes involved in carbohydrate and amino acid metabolisms. The es-sential gene sets predicted by each generic fly model were also compared with the gene essentiality dataset retrieved from the Online GEne Essentiality (OGEE) database (Gurumayum et al, 2020). We quantified the ratio of correct and incorrect essentiality pre-dictions relying on four confusion matrix categories (true positives [TP], false positives [FN], true negatives [TN], and false negatives [FN]). And, we estimated several predictive scores including sen-sitivity, specificity, accuracy, precision, F1 score, and Matthew's correlation coefficient (MCC). Although accuracy and F1 score are among the most widely favorable adopted metrics, they can be misleading in the evaluation of binary classification for particularly imbalanced datasets (e.g., many TNs but few TPs, or vice versa) (Chicco & Jurman, 2020). Yet, MCC accounts for good prediction results in all confusion matrix categories suggesting the use of this robust metric for also imbalanced datasets (Chicco & Jurman, 2020; Chicco et al, 2021). Here, we compared the predictive accuracy of the iDrosophila1, Fruitfly1, and FlySilico models considering the metrics listed in Table 2.

The sensitivity of iDrosophila1 predictions was found to be considerably higher than the other fly models. Unsurprisingly, we revealed that the FlySilico model had the worst sensitivity score because of the lack of protein complex information. In contrast to the low sensitivity values, the specificity results demonstrated that the fly models could correctly classify up to ~96–97% of the non-essential genes. Based on the precision values, the iDrosophila1

(TP: 90 and FP: 32) and Fruitfly1 (TP: 64 and FP: 25) models were shown to predict the higher ratio of correct essential genes within the predicted essential gene sets in comparison with FlySilico (TP: 3 and FP: 3). Note that six essential genes predicted by iDrosophila1 could not be included in any confusion matrix categories because of the lack of evidence, whereas 10 unclassified essential genes were identified by Fruitfly1 simulation. Because binary metrics only consider two categories, we used additional metrics based on at least three confusion matrix categories. The iDrosophila1 model was demonstrated to exhibit superior performance than the other models for these adopted metrics (accuracy, F1 score, and MCC). We found an extremely low F1 score for FlySilico predictions. It was shown to be the highest for iDrosophila1 predictions, meaning the best compromise between sensitivity and precision. One drawback of the F1 score is the tendency of this value to converge into smaller values for low sensitivity or precision. Another issue is the mis-classification potential in evaluating predictions under an imbal-anced prevalence (Hand et al, 2021). We identified the frequency of the correct nonessential genes (TN) as predominant in the pre-dicted essential/nonessential gene sets for all fly models. There-fore, we calculated MCC scores which vary in the interval of −1 and +1, indicating that a larger score reflects a better classification (Chicco & Jurman, 2020). iDrosophila1 also demonstrated a better predictive ability than the other models according to this metric. Importantly, we also performed the gene essentiality analysis in the expanded HD medium by introducing the flexible uptake rates defined in the Materials and Methods section (data not shown). In this way, 12 additional true essential genes were predicted using the iDrosophila1 model, leading to higher sensitivity (0.18) and F1 score (0.29). On the other hand, constraining the uptake rates did not change the values of the prediction metrics calculated for the Fruitfly1 and FlySilico models. Collectively, the iDrosophila1 model represented comparable or better prediction results for all metrics regardless of the uptake rates of the exchange metabolites in medium.

### Comprehensive metabolic profiling of Parkinson's disease using iDrosophila1

We further analyzed the iDrosophila1 model to evaluate the omics data-based prediction capacity of the model. In this regard, we in-vestigated differential pathways in PD, which is the second most common age-related neurodegenerative disorder worldwide (Nagoshi, 2018; Aryal & Lee, 2019). Among the PD-causing factors, mitochondrial

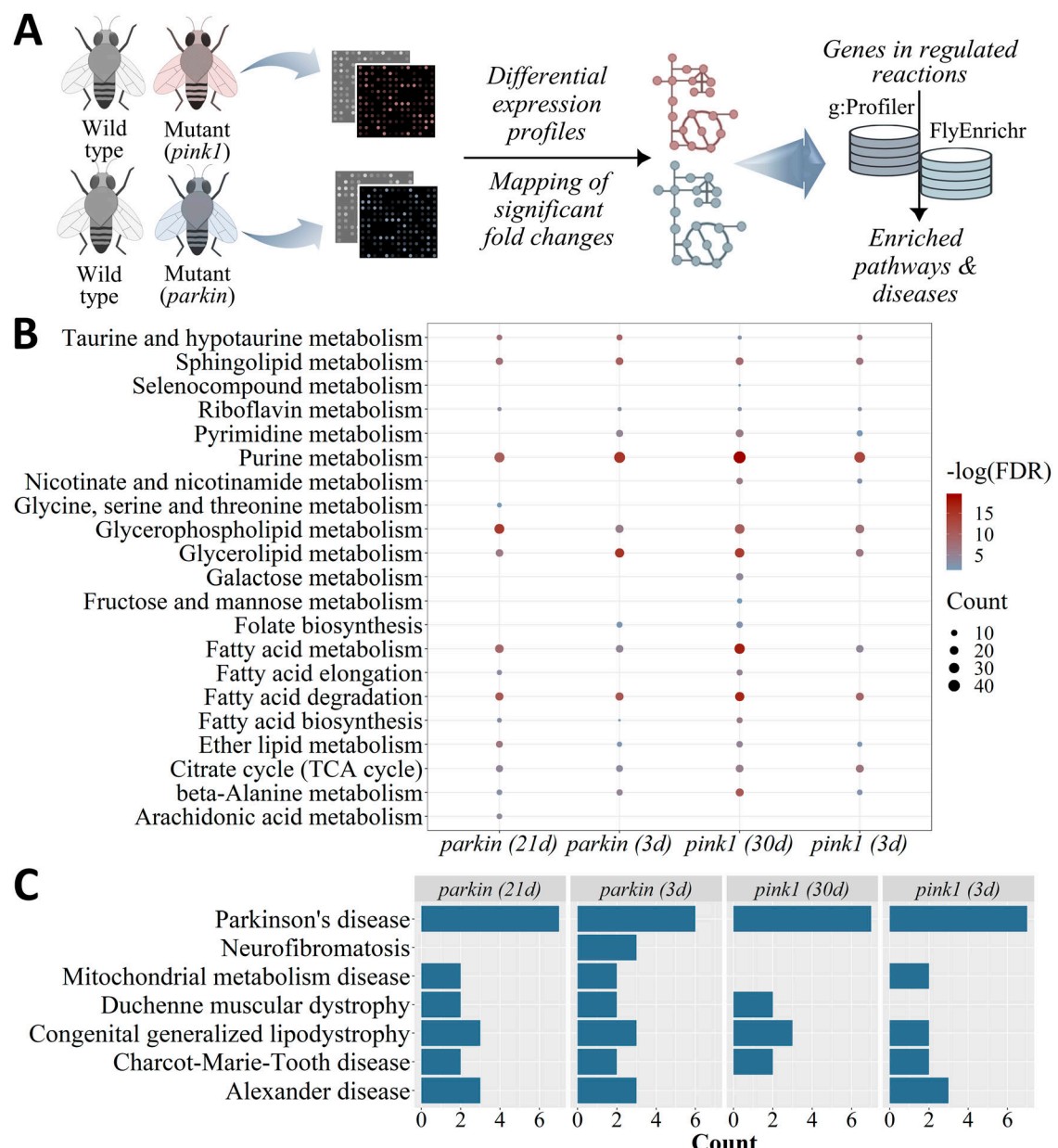

**Figure 6. Identification of enriched pathways and diseases for the predicted metabolic alterations induced by *pink1* and *parkin* mutations using the ΔFBA method.**
**(A)** Combinatory utilization of gene expression fold changes and metabolic network models are promising to explore differential reactions. **(B, C)** For the genes involved in the regulated reactions predicted by ΔFBA, (B) enriched pathways and (C) diseases are illustrated for each mutant organism in different age groups (3 d: 3-day-old, 30 d: 30-day-old, and 21 d: 21-day-old). The enriched bubble chart shows the significant KEGG pathways (vertical axis) of the regulated genes discussed in the current study. The size of the dots corresponds to the gene numbers in that pathway, and their colors represent enrichment significance. The darker red color indicates a higher significance. For the disease enrichment plot, the vertical axis indicates the enriched diseases, whereas the bars show the number of genes associated with the related disease.

dysfunction plays a pivotal role. Mitochondrial fusion and fission allow the exchange of respiratory proteins and the removal of damaged mitochondria for healthy mitochondrial homeostasis and neuro-protection (Mori et al, 1998; Gouider-khouja et al, 2003; Sasaki et al, 2004; Johansen et al, 2018). In the fission process, a reduction in mitochondrial membrane potential promotes the accumulation of PINK1 (PTEN-induced novel kinase 1) on the outer mitochondrial membrane (Imai & Hattori, 2014). Its autophosphorylation and

activation recruit Parkin (a ubiquitin E3 ligase) from the cytosol to the mitochondria. Their combined action triggers mitophagy-mediated degradation of the damaged mitochondria (Sekine & Youle, 2018), leading to a PD-like phenotype (e.g., decreased lifespan, dopaminergic neuron loss, and locomotor abnormalities) in *Drosophila* (Xu et al, 2020; Parker-Character et al, 2021).

We investigated the metabolic alterations induced by *pink1* and *parkin* mutations using transcriptomic datasets from the young and

middle-aged *Drosophila* models of PD (Celardo et al, 2017). Regulated iDrosophila1 reactions were determined via the ΔFBA method based on significant fold changes (Fig 6A). The genes involved in the regulated reactions were characterized by detecting enriched metabolic pathways (Fig 6B; also, see Table S8A–D for all enriched terms), which are either up- or down-regulated in the mutants. Fundamental pathways (amino acid, nucleotide, and lipid metabolism) were commonly overrepresented for *pink1* and *parkin* mutants. The glycine and serine metabolisms are especially prominent because of their role in the de novo nucleotide biosynthesis pathway, which is the primary strategy to sufficiently provide necessary DNA precursors for proliferating cells (Legent et al, 2006; Holland et al, 2011; Tufi et al, 2014). In this process, a variety of substrates (glycine, glutamine, and 10-formyl-tetrahydrofolate) are used. Hence, folate supplementation exhibits a stimulatory effect in de novo nucleotide biosynthesis and it has a protective role in the reduction of mitochondrial defects (Tufi et al, 2014; Celardo et al, 2017; Villa et al, 2019). We determined "purine and pyrimidine metabolism" and "folate biosynthesis" among the enriched KEGG pathways for both *pink1* and *parkin* mutants in different age groups. Rosario and colleagues showed a significant decrease in folate production in PD patients, correlated with reduced bacterial folate biosynthesis (Rosario et al, 2021). A folate-supplemented diet was reported to be useful to prevent the loss of dopaminergic neurons in *pink1* mutant flies (Tufi et al, 2014). "Riboflavin metabolism" was also overrepresented for all mutant groups (Fig 6B and Table S8), and riboflavin is required for a healthy folate cycle and energy metabolism (Wong et al, 2014). Importantly, it has a neuroprotective role by reducing glutamate excitotoxicity, oxidative stress, mitochondrial dysfunction, and NF-κB-induced neuroinflammation. Consistently, Parkin is responsible for stable glutamatergic synapses. Its mutation induces an increased susceptibility to glutamate neurotoxicity, which is linked to the onset of PD (Marashly & Bohlega, 2017). This may be evidence of the connection between riboflavin metabolism and the mitochondrial quality control process.

The "selenocompound metabolism" was enriched for the 30-d-old *pink1* mutant (Fig 6B and Table S8C). The essential micronutrient selenium induces neuroprotection by coping with oxidative stress and inflammation and shaping the gut microbiota. An increased amount of *Akkermansia* was reported in rodents via selenium supplementation. This bacterium is involved in gut barrier protection, immune modulation, and the regulation of host metabolism (Arias-Borrego et al, 2019; Callejón-Leblic et al, 2021). Selenium shows an age-dependent decline in humans, and nutraceuticals and selenium-enriched functional foods have received a growing interest in recent years (Arias-Borrego et al, 2019; Callejón-Leblic et al, 2021). Another enriched metabolic process related to neuroprotection is "β-alanine metabolism" (Fig 6B and Table S8). This nonproteinogenic amino acid is a precursor of carnosine dipeptide including diverse functions (e.g., proton buffering, metal chelation, antioxidation, and muscle contractility). In PD patients, the supplementation of carnosine dipeptide showed its therapeutic potential to improve impaired motor activity (Rezende et al, 2020). Consistently, there is a relationship between the altered level of β-alanine and PD physiopathology (Solana-manrique et al, 2022). More specifically, this compound contributes

to the enhanced levels of extracellular GABA and dopamine in substantia nigra (Allman et al, 2018). In addition, taurine and β-alanine supplementation were shown to support muscle function in mice by increasing fatigue resistance in muscles and reducing contraction-induced oxidative stress (Horvath et al, 2016). Taurine is another β-amino acid enriched in our results, and it is involved in neuromodulation, $Ca^{2+}$ homeostasis, and the regulation of antioxidant processes (Fig 6B and Table S8). At a high concentration in the substantia nigra, it can regulate dopamine release and dopaminergic neuron activity. "Nicotinate and nicotinamide metabolism" was also shown to be enriched for *pink1* mutant flies (Fig 6B; also, see Tables S8A and C). Recently, nicotinamide riboside was reported to have a neuroprotective effect. It regulates gene levels associated with oxidative stress response, mitochondrial respiration, inflammatory response, histone acetylation, and proteasomal metabolism by elevating cerebral NAD levels in PD patients (Brakedal et al, 2022).

Lipid metabolic pathways including "fatty acid metabolism," "fatty acid degradation," "glycerolipid metabolism," "glycerophospholipid metabolism" "sphingolipid metabolism," "ether lipid metabolism," and "arachidonic acid metabolism" were overrepresented with the genes that control the altered reactions in *pink1* and *parkin* mutants (Fig 6B and Table S8). The lipid composition of synaptic vesicle membranes regulates their interactions with α-synuclein (α-syn) proteins, whose aggregation is the main pathological hallmark of PD (Giguère et al, 2018; Chia et al, 2020; Mori et al, 2020). Iljina et al reported the protective role of arachidonic acid against the production of toxic beta-sheet structures. Arachidonic acid is known to support the formation of alpha-helical-folded multimers of α-syn with higher resistance to fibril formation (Iljina et al, 2016). Furthermore, the accumulated PINK1 and Parkin proteins at endoplasmic reticulum–mitochondria contact sites regulate inter-organelle communication related to lipid metabolism, $Ca^{2+}$ signaling, and mitophagy (Gómez-Suaga et al, 2018; Barazzuol et al, 2020; Fais et al, 2021). Overall, mitochondrial activity is heavily linked to lipid metabolism, and the altered lipidomic composition of mitochondrial membranes in Parkin knock-out mice was documented (Fais et al, 2021). In addition, fatty acid oxidation contributes to acetyl-CoA production that is metabolized in the tricarboxylic acid cycle. The down-regulation of many tricarboxylic acid compounds was shown in the *pink1* mutant flies (Tufi et al, 2014). Similarly, we showed altered tricarboxylic acid cycle and carbon metabolism in the *pink1* and *parkin* mutants (Fig 6B and Table S8). In agreement with these results, the *pink1* mutation causes the reprogramming of glucose metabolism to assist metabolic adaptation (Requejo-Aguilar et al, 2014). In addition, the fructose level was shown to increase in PD patients suggesting its potential protective role against oxidative stress through respiratory shifting in the early stages of the disease. Carbon sources also mediate protein modifications by glycation or glycosylation (Videira & Castro-Caldas, 2018). We found the enrichment of metabolic pathways linked with several glycating agents (e.g., fructose, galactose, and mannose) in 30-d-old *pink1* mutant flies (Fig 6B and Table S8B).

Lastly, we determined diseases significantly enriched with the genes involved in the regulated reactions by *pink1* and *parkin* mutations. To do so, we compared the gene list with the gene set library in FlyEnrichr derived from "Human Disease data from

FlyBase (2017)," which provides disease–gene association information. Neurogenerative disorders, muscular diseases, and mitochondrial diseases were found to be overrepresented with the genes we identified, further confirming the accuracy of transcriptome-guided iDrosophila1 predictions (Fig 6C). Of the enriched disorders, "Duchenne muscular dystrophy" is characterized by mitochondrial dysfunction and impaired mitophagy. Increased inflammation occurs because of defective mitochondria, and it further exacerbates disease pathology (muscle damage and increased fibrosis) (Reid & Alexander, 2021). Similarly, Charcot–Marie–Tooth disease leading to severe muscular deficits is related to defective mitochondrial processes similar to "mitochondrial metabolism disease" and "Parkinson's disease" (Fig 6C) (Nandini et al, 2019; Schiavon et al, 2021). Together, these preliminary findings are remarkable to indicate the capacity of iDrosophila1 in the discovery of context-specific pivotal metabolic alterations. Thus, this analysis further supported our model predictions by suggesting that iDrosophila1 may be useful to gather insight into complicated human diseases.

In conclusion, there is still a gap in the use of this organism for comprehensive condition-specific metabolic modeling although *Drosophila* is a workhorse in experimental studies. Given the increasing amounts of multi-omics data, the iDrosophila1 models contextualized by diverse disease-specific omics datasets may further contribute to systems medicine. Such models may provide novel insights for human disorders and biomarker detection. In addition, community models are currently used to represent cell populations in single/multiple tissue(s), whole body, microbial communities, and host–microbial group interactions. Considering the dramatic impact of the microbiota on human health and the low microbial diversity of *Drosophila*, our metabolic network model can also be useful to elucidate metabolic interactions between *Drosophila* and gut microbiota. Overall, iDrosophila1 may provide avenues to advance fly metabolic modeling and better understand more complicated human metabolism.

## Materials and Methods

### Metabolic reconstruction procedure

We used an orthology-based approach to reconstruct a draft *D. melanogaster* model using a flux-consistent version of the HMR2 model (7,518 reactions, 5,426 metabolites, and 2,479 genes) as a template (Radic Shechter et al, 2021; Zirngibl, 2021). We curated the GPR associations of the template model by improving the protein complex information based on another human model (iHsa), which includes an extensive manual refinement of the GPR rules of HMR2, particularly considering the definition of enzyme complexes (Blais et al, 2017). The orthology-based inference of *Drosophila* model was achieved by replacing the genes in the GPR rules of the template model with *Drosophila* orthologs. To do so, "human orthologs" list of *D. melanogaster* was retrieved from the FlyBase database (last accessed: 02/02/2021) (Larkin et al, 2021). The list includes information about evidence scores (DIOPT scores), *Drosophila* genes (FlyBase gene IDs), their human orthologs, and associated disease phenotypes. The DIOPT score refers to the number of tools supporting a given orthology prediction (Hu et al, 2011). It was used in

orthology-based gene mapping as a scoring approach (3–15) to increase the confidence level (Table S1). Two criteria were considered in this approach: (1) if a human gene has multiple *Drosophila* orthologs with different DIOPT scores, only the ortholog(s) with maximum DIOPT scores were considered; (2) when a human gene has over nine *Drosophila* orthologs, this gene association was ignored (Blais et al, 2017) because the orthology association was not specific. However, the related reactions were kept in the model because the existence of *Drosophila* orthologs showed that the corresponding reaction is available in *Drosophila*. Lastly, the GPR rules of the template model were modified by replacing human genes with their *Drosophila* orthologs at a high confidence level, and the human genes that could not be matched with any *Drosophila* orthologs were discarded from the model. On the other hand, all non-enzymatic reactions in the template model were included in the draft model.

To enhance gene coverage of the draft model, we created another metabolic network for *D. melanogaster* using the RAVEN toolbox (Wang et al, 2018) by considering protein homology against KEGG (Kanehisa & Goto, 2000) and MetaCyc (Caspi et al, 2012) databases. Based on the study of Wang et al (2018), we first reconstructed a combined KEGG-based metabolic network by merging two different KEGG-based networks: (1) we used KEGG organism identifier (dme) for the first network only, and (2) the second network was reconstructed using the BLAST algorithm to query the protein sequence of *Drosophila* from the FlyBase database (version number: r6.37) against the latest pre-trained hidden Markov model (euk100_kegg94). The same parameters (default cut-off: $10^{-50}$, minScoreRatioG: 0.95, and minScoreRatioKO: 0.7) were used in the reconstruction of both KEGG-based networks. Incomplete information and undefined stoichiometries were excluded from the network. In addition to the combined KEGG-based metabolic network, we generated a MetaCyc-based metabolic network using another automatic reconstruction function in the RAVEN toolbox. In the reconstruction process, default parameters (bit-score ≥ 100 and positives ≥ 45%) were chosen, and unbalanced/undetermined reactions were excluded from the network.

The KEGG- and MetaCyc-based metabolic networks were subsequently merged (hereafter referred to as the KEGG–MetaCyc network). The genes that were not shared with the HMR2-based draft model were identified. These KEGG–MetaCyc-specific genes were functionally characterized by identifying significantly enriched KEGG pathways using the g:Profiler web server (Raudvere et al, 2019) with a false discovery rate (FDR) of 0.05 (Table S2). The reactions associated with the KEGG–MetaCyc-specific genes were selected and included in the draft model. Because of the lack of compartmentalization in the KEGG–MetaCyc network, subcellular protein localizations from several sources were used to assign at least one compartment for each KEGG–MetaCyc-specific gene (Table S3). The sources covered in this study include COMPARTMENTS (Binder et al, 2014), FlyBase (Larkin et al, 2021), GLAD (Hu et al, 2015), QuickGO (Huntley et al, 2009), AmiGO 2 (The Gene Ontology Consortium, 2015), UniProt (Uniprot Consortium, 2021), Reactome (Fabregat et al, 2016), CELLO2GO web server (Yu et al, 2014), and a mass spectrometry-based study (Tan et al, 2009). It is important to note that gene ID consistency is crucial for accurate gene–compartment mapping. Therefore, the FlyBase ID Validator tool

(Larkin et al, 2021) was used to convert the gene IDs from the subcellular localization databases and tools to the current versions of FlyBase gene IDs. A compartment dictionary was then created by mapping the compartment information to the corresponding KEGG–MetaCyc-specific genes based on the Gene Ontology (GO) cellular component terms. Only the compartments found in the draft model were considered in the construction of this dictionary. After the compartmentalization of the genes, the compartment dictionary facilitated the transfer of compartment information from the genes to the KEGG–MetaCyc reactions based on their GPR associations.

The HMR2-based draft *Drosophila* model was merged with the compartmentalized KEGG–MetaCyc-specific model by adding each KEGG–MetaCyc reaction to the draft model using the *addReaction* function in the COBRA toolbox (Heirendt et al, 2019). The biomass formation equation derived from the template human model was curated, and then we identified biomass components that could not be synthesized by the merged model using the COBRA *biomassPrecursorCheck* function. Then, the gap-filling algorithm in the RAVEN toolbox, *fillGaps*, was used to add reactions required for the synthesis of these biomass components. To ensure the production of all biomass components, the lower bound of the biomass formation reaction was set to 0.1 while running the *fillGaps* algorithm. We also set the "useModelConstraints" parameter in the *fillGaps* to true. In the gap-filling process, the template human model was used as a reference to fill in the missing knowledge about metabolism. First, the reference model and merged model were set to a chemically defined medium with unlimited uptake rates (1,000 mmol g$^{-1}$h$^{-1}$). This growth condition was termed an "expanded holidic diet (HD)" because of the addition of several vitamin derivatives to the HD medium (see Table S5) (Piper et al, 2014; Schönborn et al, 2019). Subsequently, new reactions were added to the merged model by gap filling from the human model without gene information (Table S4).

After the gap-filling step, cholesterol metabolism was revised to represent cholesterol auxotrophy in *Drosophila* (Table 1). In addition, the GPR rules were updated based on the information on 556 *Drosophila* protein complexes reported by Guruharsha and colleagues (2011) (Guruharsha et al, 2011). Additional curations were introduced for the GPR rules based on the FlyBase gene group list and the literature (Van den Berghe et al, 1997; Santos & Lehmann, 2004; Allan et al, 2005; Wahl et al, 2005; Avval & Holmgren, 2009; Grant et al, 2010; Pavlovic & Bakovic, 2013; Tang & Zhou, 2013; Kemppainen et al, 2014; Attrill et al, 2016; Kovacs et al, 2018; Marygold et al, 2020b; Rhooms et al, 2020). We also investigated the presence of leaking energy metabolites in the draft model. These metabolites represent the compounds that can be produced even in the absence of any nutritional intake. For instance, leaking ATP metabolite can lead to spontaneous energy production without any nutrient uptake, and it can result in a higher growth rate. In the leak testing, we analyzed the metabolism of the charging energy metabolites listed by Fritzemeier and colleagues (e.g., ATP, NADH, NADPH, FADH$_2$, GTP, and H$^+$) (Fritzemeier et al, 2017). To this end, we first constrained all uptake reactions in the model to zero by allowing only their secretions. Then, a dissipation reaction was added for each energy metabolite if the corresponding reaction was not available in the model. Maximization of the rate of each dissipation reaction

was defined as an objective function to evaluate whether the corresponding metabolite is leaking or not. A nonzero rate for the dissipation reaction indicated the leakage of that metabolite. To overcome this problem, we performed double-reaction deletions by setting their lower and upper boundaries to zero under the same nutritional condition, where all the uptake reactions were constrained to zero flux. The reaction pairs whose deletion prevented the production of a leaking metabolite in the absence of nutrients were carefully revised by removing incorrectly compartmentalized or redundant reactions. It is also worth emphasizing that the reaction "HMR_4762" mediating porphyrin metabolism was updated to revise the incorrect link between heme and cytochrome-C metabolites. This reaction is catalyzed by cytochrome-C heme lyase (FBgn0038925) and provides the conversion of apocytochrome-C and heme to cytochrome-C in a reversible manner. Because of the incorrect conversion of heme to the cytochrome-C in the template human model, we first added apocytochrome-C to the HMR_4762 in the *Drosophila* model for a proper metabolic conversion. Then, the missing reactions associated with the apocytochrome-C metabolism were added from the HMR2 model. These changes affect the growth rate because of the presence of cytochrome-C in the 'cofactor and vitamin' composition of the biomass equation. It should be noted that we performed several additional curation steps for all components of the metabolic networks (reactions, metabolites, and genes) considered in the reconstruction steps of the *Drosophila* model. The details of these steps are given in the next section.

The final *Drosophila* model was called iDrosophila1, and it is available in our GitHub repository (https://github.com/SysBioGTU/iDrosophila). The metabolites, reactions, and genes in the model were annotated using a variety of sources (HMR 2.0, Metabolic Atlas [Pornputtapong et al, 2015], FlyBase, MetaNetX [Moretti et al, 2021], ModelSEED [Seaver et al, 2021], BiGG Models [King et al, 2016], Metabolite Translation Service, MBROLE 2.0, MetaboAnalyst, Chemical Translation Service, MetaBridge [Hinshaw et al, 2018], and SBOannotator [Leonidou et al, 2023]). The model can be accessible in MATLAB (MAT), XML (SBML), and JSON formats, which are compatible with the COBRA Toolbox model structure. The quality of the iDrosophila1 was evaluated via the metabolic model test suite, MEMOTE (version 0.13.0) using the Gurobi solver (Gurobi Optimization, LLC).

## Curation of reaction, metabolite, and gene components

Reaction-centric, metabolite-centric, and gene-centric curations were commonly applied to the metabolic networks in the reconstruction steps as detailed below. This covers the curation of metabolic redundancy, name standardization, and the removal of missing/incomplete model components.

### Reaction-centric curations

Reaction-centric curation included the detection and removal of duplicated reactions, which arose because of the merging of two metabolic models; KEGG–MetaCyc-specific model and template human model (Radic Shechter et al, 2021; Zirngibl, 2021). The duplicated reactions were identified iteratively by ignoring common currency metabolites (H$^+$, H$_2$O, and P$_i$) and compartment information. If there is any compartmental inconsistency between the duplicated reactions with the same gene content, the compartment information

in the template human model was accepted to be correct. This step was performed in a tightly coupled manner with the curation of GPR rules for all models given in the current study. If the duplicated reactions are derived from different metabolic networks, the GPR rules of these reactions were manually examined and combined before the removal of one reaction by avoiding any gene loss. In addition, trivial or incomplete reactions with missing stoichiometric coefficients were removed from the models.

### Gene-centric curations

Gene-centric curations allowed both standardization of gene names and the elimination of gene-based redundancy in models. HMR2-based draft *Drosophila* model was reconstructed using FlyBase gene IDs. On the other hand, KEGG- and MetaCyc-based networks include FlyBase protein and annotation IDs. Therefore, all genes were denoted based on the current versions of FlyBase gene IDs for compatibility between the models. However, this led to the emergence of redundant genes in a given GPR association. Similarly, the conversion of Ensembl gene IDs into the FlyBase gene IDs in the HMR2-based reconstruction caused the emergence of redundant genes in a given GPR association because of the presence of multiple *Drosophila* orthologs for some human genes. The redundant genes/ gene associations in each GPR rule were manually curated.

### Metabolite-centric curations

Metabolite-centric curations are based on the elimination of trivial and synonymous metabolites. The trivial metabolites, which were not associated with any reactions (no assigned stoichiometric coefficients), were investigated in all models, and removed. In addition, synonymous metabolites were identified to eliminate metabolic redundancy.

First, synonymous metabolites within the model were interrogated based on compound names (metabolite name and BioCyc name) and IDs (KEGG [Kanehisa & Goto, 2000], ChEBI [Degtyarenko et al, 2008], PubChem [Kim et al, 2021], and LIPID MAPS [Fahy et al, 2007]). To this aim, the compound information was collected from available metabolic networks and several web servers including Metabolite Translation Service (Caspi et al, 2016), MBROLE 2.0 (López-Ibáñez et al, 2016), MetaboAnalyst (Chong et al, 2018), and Chemical Translation Service (Wohlgemuth et al, 2010). Note that only three-star ChEBI IDs (parent IDs) that were manually curated by the ChEBI team were kept in the compound list by excluding less reliable ChEBI IDs. Using this list, we subsequently generated a paired compound name and ID (collectively called identifiers) dictionary by mapping the related IDs to the metabolites in the given model. Based on these identifiers, we matched the model metabolites by three main criteria: (1) if two metabolites within the model have at least one common identifier and they do not have any common identifier with any other metabolites, they were accepted as synonyms; (2) when three or more metabolites were detected to have common identifiers, a filtering step was applied by assuming that the metabolite pair with the maximum number of matched identifiers (i.e., the highest number of matched IDs) were synonymous; (3) in the case where more than two metabolites have common identifiers, there can be more than one pair of metabolites with the maximum number of identifiers. Such hits were further investigated to select the most possible synonymous

metabolites with extensive knowledge-based manual curations based on the databases including BioCyc (Karp et al, 2018), ModelSEED (Seaver et al, 2021), ChemSpider (Pence & Williams, 2010), and MetaNetX (Moretti et al, 2021) in addition to the literature (Fig S1). For each metabolic match, one of the duplicated metabolites was removed from the model after the assembly of their stoichiometric coefficients in the model. This curation step was separately performed for the template human model, HMR2-based draft *Drosophila* model, and KEGG–MetaCyc-specific network.

Second, we uncovered synonymous metabolites across the models to successfully merge the networks. This step was employed before combining the HMR2-based and KEGG–MetaCyc-specific metabolic networks to prevent any metabolic redundancy in the merged model. Similar to the first approach detailed above, a compound name–ID pair dictionary was generated for each model. Then, the metabolites in both models were mutually matched based on the specified identifiers. The same assumption was used to prioritize the multi-matched metabolites (Fig S1). After the manual confirmation of the potential synonyms, the names of synonymous metabolites in the KEGG–MetaCyc-specific network were replaced with the metabolite names in the HMR2-based draft *Drosophila* model.

### Curation of biomass formation reaction

The biomass equation of the draft *Drosophila* model derived from the reference human model was updated based on the FlySilico model (Schönborn et al, 2019) and the literature. Biomass composition was examined comparatively with the FlySilico, and most metabolites were found to be common (amino acids, glycogen, triglyceride, and cholesterol). The missing growth-associated ATP maintenance reaction was included in the model using the stoichiometric coefficients in the FlySilico model. Furthermore, cardiolipin was removed from the biomass reaction because it was reported to be nonessential in *D. melanogaster* (Kubota-sakashita, 2020). Vitamin D was also removed from the biomass reaction because fruit fly is known as a cholesterol auxotroph, and 7-dehydrocholesterol (a cholesterol precursor) synthesis is an overlapping reaction in cholesterol and vitamin D synthesis. The gene encoding the enzyme responsible for the synthesis of 7-dehydrocholesterol was reported to be absent in the fly genome (Santos & Lehmann, 2004).

### Validation of growth phenotypes

iDrosophila1 model was checked to ensure the production of all biomass precursors. After ensuring that the growth rate of the network was nonzero, we analyzed cholesterol auxotrophy, aspartate nonessentiality, and the impact of essential amino acids under the expanded HD condition (Table S5). Here, we allowed the flexible intake of 47 dietary compounds by limiting the maximum sucrose uptake rate to ~ 2.212 mmol $g^{-1}h^{-1}$ (Schönborn et al, 2019). The maximum vitamin uptake rates were set to 1/100 of the sucrose uptake rate because of the low vitamin consumption tendency of organisms, whereas the use of remaining HD substances (except for salts and water) was set to 1/10 of the sucrose uptake rate. We

constrained the maximum oxygen uptake rate to 24 mmol $g^{-1}h^{-1}$ by estimating the oxygen level required to consume all sucrose through aerobic respiration. Under this condition, we investigated the effect of increasing cholesterol and amino acid levels on growth rate as explained in the study of Schönborn et al (2019). To predict the growth rates, maximum biomass production was defined as the objective function, and the FBA approach (Orth et al, 2010) with the Gurobi solver was used to identify intracellular flux distributions.

**Validation of essential gene predictions**

To discover vital genes in the iDrosophila1 model, gene essentiality analysis was performed under the expanded HD condition by allowing an infinite intake of all diet compounds. Each gene was deleted by suppressing the corresponding reactions, and FBA was performed with the objective of growth maximization with the Gurobi solver. This step was achieved using the *singleGeneDeletion* function in the COBRA toolbox (Schellenberger et al, 2011). The effect of each single-gene knockout on the biomass formation was assessed based on the specified cutoff value. To evaluate gene essentiality, different cutoff values were preferred in the previous studies for prokaryotic and eukaryotic metabolic models (Pratapa et al, 2015; Khodaee et al, 2020; Wang et al, 2021). Khodaee et al determined the essential genes in their mouse models using 30% of the optimal growth rate as a threshold value (Khodaee et al, 2020). We used the same cut-off of 30% to predict essential genes. In this regard, when the deletion of a gene resulted in a significantly reduced growth rate (i.e., a smaller growth rate than the given cutoff), this gene was considered essential. In this way, we uncovered essential and nonessential gene sets. The genes involved only in inactive (blocked) reactions were subsequently discarded from the list of gene sets because they do not affect biomass formation. The blocked reactions were identified using flux variability analysis (Mahadevan & Schilling, 2003) under the expanded HD condition. If the sum of absolute minimum and maximum fluxes of a reaction was less than $10^{-5}$ in flux variability analysis, it was accepted as inactive. After determining essential and nonessential gene sets in the active reactions, we characterized the essential genes (Table S6A) through the identification of enriched biological processes (Table S6B) and KEGG pathways (Table S6C) using the g:Profiler web server for FDR at the 0.05 level. In the next step, we assessed the predictive capability of the model based on the experimental gene essentiality dataset, which was stored in the OGEE database (Gurumayum et al, 2020). In this dataset, we classified conditionally essential genes as essential. Finally, several metrics (sensitivity, specificity, accuracy, precision, F1 score, and MCC) were calculated to evaluate the model performance based on the OGEE dataset.

**iDrosophila1-mediated analysis of differential metabolic pathways in Parkinson's disease**

We further evaluated the capacity of the iDrosophila1 model in phenotypic predictions through the investigation of age- and PD-dependent differential pathways in *D. melanogaster*. In this process, we used a microarray dataset (Agilent) from ArrayExpress (accession number: E-MTAB-1406) (Celardo et al, 2017). It includes the samples from the heads of male flies harboring *pink1*[B9] (*pink1*) and *park*[25] (*parkin*) mutations at different age groups: young flies (3-d-old) and middle-aged flies (21-d-old [*parkin*] and 30-d-old [*pink1*]). For each age group, there are three and six biological replicates for *pink1* and *parkin* mutant flies, respectively. The Limma package (Law et al, 2014) for R version 4.1.0 was used to process, normalize, and analyze the data. In the data-processing step, the effects of nonspecific signals in the dataset were removed using the *backgroundCorrect* function after reading the intensity data via the *read.maimages* function. Then, the *normalizeBetweenArrays* function was used to establish consistency between different arrays. Using this normalized log-transformed dataset, differential expression analysis was performed to uncover significant fold change values for each mutant group relative to the corresponding control group (age-matched WT fly). To ensure the consistency of the genes with the iDrosophila1 model, all gene annotation IDs were converted to the current versions of FlyBase gene IDs using the FlyBase ID Validator tool (Larkin et al, 2021). In this curation step, multiple hits were manually revised. The limma-trend function was subsequently used by setting robust = TRUE. Genes with significant changes in their expression levels (*P*-value < 0.01) were selected for further analysis. Based on the fold change cutoff values, we applied another filtering process for the genes with multiple probe measurements using the following criteria: (1) if one or more probe(s) of a gene have fold changes ≥1.5, the maximum of fold change values of its probes was considered by assuming the up-regulation of this gene; (2) when the fold change values of any probe(s) ≤ 0.67 (~1/1.5) for a gene, the minimum fold change value was assigned by assuming the down-regulation of this gene (3) if all probes of a gene had moderate (0.67 < fold change < 1.5) or ambiguous fold change values (i.e., the presence of both up-regulated and down-regulated probes), the average fold change was assigned to this gene.

In the metabolic network analysis step, the maximum uptake rates of all exchange metabolites were set to unrestricted flux (i.e., 1,000 mmol $g^{-1}h^{-1}$). Then, the filtered fold change values were mapped to the reactions in the iDrosophila1 model using the COBRA function *mapExpressionToReactions*. GPR associations were taken into consideration in the mapping process. The minimum fold change value was assigned to the reactions whose corresponding genes are linked with "AND" operator whereas the maximum fold change was used for the genes that are linked with "OR" operator. Differential reaction expression levels were then used to elucidate PD-induced altered metabolism via a recent approach, ΔFBA with default parameters (Ravi & Gunawan, 2021). Note that the flux of the non-growth-associated ATP maintenance was assumed to be unchanged between WT and mutant groups. The ΔFBA algorithm calculates flux changes (Δv) between two diverse conditions by applying a two-step optimization procedure: it maximizes consistency and minimizes inconsistency between Δv and differential reaction expressions. Based on the predicted Δv distribution, we identified the altered (up-regulated and down-regulated) reaction sets with differential fluxes above the specified threshold ($|\Delta v_i| > 0.1\%$ of the largest flux bound). GPR rules were used to determine the corresponding genes involved in the regulated

reactions. These genes were characterized in terms of significantly enriched KEGG pathways (FDR < 0.05) and diseases (*P*-value < 0.01) using the g:Profiler (Raudvere et al, 2019) and FlyEnrichr (Kuleshov et al, 2016) web servers.

## Data Availability

The final iDrosophila1 model has been deposited in MAT, SBML, and JSON formats in GitHub: https://github.com/SysBioGTU/iDrosophila.

## Supplementary Information

## Acknowledgements

We would like to thank Katharina Zirngibl for providing the genome-scale metabolic network of human, which was used as the template in the current study.

### Author Contributions

MF Cesur: formal analysis, investigation, methodology, and writing—original draft, review, and editing.
A Basile: methodology and writing—review and editing.
KR Patil: conceptualization, supervision, methodology, and writing—original draft, review, and editing.
T Çakır: conceptualization, supervision, methodology, and writing—original draft, review, and editing.

### Conflict of Interest Statement

The authors declare that they have no conflict of interest.

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
