## [Reviewer comments · Life Science Alliance]

Life Science Alliance

A new metabolic model of *Drosophila melanogaster* and the integrative analysis of Parkinson's disease

Müberra Cesur, Arianna Basile, Kiran Patil, and Tunahan Çakır

DOI: <https://doi.org/10.26508/lsa.202201695>

Corresponding author(s): Tunahan Çakır, Gebze Technical University

Review Timeline:

Submission Date:	2022-08-27
Editorial Decision:	2022-10-13
Revision Received:	2023-03-31
Editorial Decision:	2023-05-02
Revision Received:	2023-05-11
Accepted:	2023-05-12

Scientific Editor: Novella Guidi

Transaction Report:

October 13, 2022

Re: Life Science Alliance manuscript #LSA-2022-01695-T

Tunahan Çakır
Gebze Technical University
Turkey

Dear Dr. Çakır,

Thank you for submitting your manuscript entitled "A genome-scale metabolic model of *Drosophila melanogaster* for integrative analysis of brain diseases" to Life Science Alliance. The manuscript was assessed by expert reviewers, whose comments are appended to this letter. We invite you to submit a revised manuscript addressing the Reviewer comments.

Thank you for this interesting contribution to Life Science Alliance. We are looking forward to receiving your revised manuscript.

Sincerely,

B. MANUSCRIPT ORGANIZATION AND FORMATTING:

Reviewer #1 (Comments to the Authors (Required)):

In the present manuscript, Cesur and coworkers present a new genome-scale metabolic network reconstruction of *Drosophila melanogaster*, to facilitate the study of metabolism in this organism, which may give new insights into important biological processes (including development and neurodegenerative diseases). This reconstruction, to my knowledge, is the most comprehensive manually-curated genome-scale reconstruction for the metabolic network of fruit fly, which also includes cell compartmentalization. I can recommend this manuscript for publication in case the authors address the following issues.

1. It is not completely clear to me how the authors investigate (and then, fix the problem of) the metabolic leaks. Firstly, the metabolic leaks are presumably a result of the presence of mass-unbalanced reactions (or, the so-called metabolic inconsistencies; see: DOI: 10.1093/bioinformatics/btn425). Did the authors find such inconsistencies in the network? Secondly, the authors mention that "To overcome this issue, we performed multiple reaction deletions and identified the reactions causing metabolic leaks. The identified reactions were manually curated". To me, this procedure is rather ambiguous. I strongly recommend the authors provide details for this procedure.
2. I strongly recommend the authors use the MEMOTE server to validate their model, (and possibly, report the qualification results as a supplementary file).
3. In line 110, it is mentioned that "If a human gene has over nine *Drosophila* orthologs, this gene association was ignored". Please elaborate on the reason behind this step.
4. Please explain the importance of aspartate in fruit fly metabolism (line 210). Why this specific amino acid is studied in this work (but not the other amino acids)?
5. About the study of gene knockouts (lines 226-228) there are at least two important issues. At first, a cutoff of <1% for inferring the lethality of a gene makes sense in the case of microorganisms, but not necessarily in the case of complex multicellular organisms like *D. melanogaster*. Secondly, the network reconstruction here represents the "generic" metabolism of fruit fly, while essentiality is a cell-type specific concept. I recommend the authors to, at least, mention these issues as the potential pitfalls of their approach to studying gene essentiality.
6. To me, it is not completely clear how metabolite names and IDs are matched. Supplementary figure 1 also provides incomplete information on this issue. Please provide further details.

Reviewer #2 (Comments to the Authors (Required)):

This paper presents a new, thorough and comprehensive metabolic network model for *Drosophila melanogaster*. The first two sections of the Results describe how the new model was derived through computational and manual curation steps. The following sections validate and apply the model with respect to: i) the effect of increasing cholesterol/aspartate levels; (ii) gene essentiality; and (iii) metabolic phenotypes associated with Parkinson's disease. Overall, this is an advance on previous published models and is a welcome addition to the resources available to researchers of *Drosophila* metabolism. I judge the paper to be suitable for publication after several points have been addressed, most notably a clearer account of which parts of the model are based on computational (orthology-based) predictions versus direct evidence from *Drosophila*, and a clearer statement about how the computational model can be accessed/used. (I will note that I am not an expert in constructing/validating metabolic models so I cannot critique those technical aspects.)

Major points to address:

1. The current title suggests the metabolic model is specific/restricted to the neural system, which is not the case and thereby undersells the scope and applicability of the model. Suggest the last 6 words of the title are either omitted or modified.
2. As far as I can tell, the only way for researchers to access/use the iDrosophila1 model is via the GitHub repository that is mentioned discretely at the end of Methods section 2.1. This availability statement needs to be much more prominent - ie. mentioned in the Abstract and/or Results section. It would also be useful to include the following sentence from the Github site within the manuscript: "The model is shared in MATLAB .mat, .xml (SBML), and .json formats, which are compatible with COBRA Toolbox model structure".
3. The primary *Drosophila* model is based on a human metabolic model called 'HMR2'. Assuming there are multiple human models available, can the authors say why they chose to use this one? E.g. is it the most recent, or the most comprehensive, or perceived as the 'best' for some reason? A few more details about HMR2 would also be appreciated - is the model based

entirely on empirical data from human cells, or are aspects of it computed from studies on other organisms?

4. In several instances, the main text appears to suggest that the KEGG-MetaCyc model is not an orthology-based model, but is bringing in *Drosophila*-specific, perhaps empirical, information. E.g. in the abstract: "The gene coverage and metabolic information of the orthology-based draft model were expanded via *Drosophila*-specific KEGG and MetaCyc databases"; or from the introduction: "we first developed a draft model through the orthology-based mapping of *Drosophila* genes. Then, the draft *Drosophila* model was expanded based on metabolic information in KEGG and MetaCyc databases"; or from the second paragraph of section 3.1: "Using metabolic information in KEGG and MetaCyc databases, a KEGG-MetaCyc metabolic network was also generated for *D. melanogaster*". But the Methods section says that the authors "generated another metabolic network for *D. melanogaster* using RAVEN toolbox [21] by considering protein homology against KEGG [22] and MetaCyc [23] databases". Thus, the KEGG-MetaCyc model was generated by a similar method to the "orthology-based model", in that gene-protein-reaction rules from a curated set were transferred to the *Drosophila* genome based on sequence similarity in both cases. I recommend that: (i) the two models are referred to as the 'HMR2-based model' and the 'KEGG-MetaCyc model', to avoid distinguished the former as the only 'orthology-based model'; and (ii) the authors provide a clearer description of which reactions/pathways in the final i*Drosophila*1 model are based purely on orthology/sequence similarity, versus any that they manually curated from the *Drosophila* literature and thus have direct experimental evidence.

5. In section 3.5 (and Figure 6B), it's unclear whether the genes/pathways identified as 'enriched' are up- or down-regulated in the in the parkin or pink1 mutants. Please clarify.

6. The last paragraph of section 3.5 describes analysis of "enriched diseases associated with [*Drosophila*] pink1 and parkin mutations" (Fig 6C). It's unclear what it means for a disease to be associated with mutations in these fly genes, or how this association was determined. I couldn't find an explanation of this in the Methods. Please clarify.

Minor points to address:

1. I think this sentence in the introduction is wrong: "The highly conserved disease pathways of human have been extensively analyzed based on the *Drosophila* genes, which share 75-77% homology with disease-associated human genes [3, 4]." The relevant sentences from the given references are "about 75% of the genes responsible for human diseases have homologs in flies" and "nearly 75% of human disease-causing genes are believed to have a functional homolog in the fly", which is a different statistic.

2. Please define what the 'i' in 'i*Drosophila*1' represents.

3. It's mentioned in section 3.1 and in the supplemental note that gene duplicates (paralogs) encoding the same enzyme were reviewed so that only one gene was present in the gene-protein-reaction and the final model. The note should mention how the 'winning gene' was chosen, and it would be useful to have a supplemental table listing these cases. Such paralogs may have important tissue or context-specific roles and are not necessarily 'redundant'.

4. It would be helpful to mention at the end of the first paragraph of section 3.1 that the HMR2-based model includes intracellular compartments, rather than only introducing this point when describing the KEGG-MetaCyc model later.

5. Section 3.1 and the supplemental note describe the need for and utility of cataloguing the different names/IDs for the same metabolite in different source databases. E.g. the supplementary note says "potential synonyms were subsequently revised via an extensive manual curation step to avoid wrong matches due to obsolete or incorrectly assigned IDs." Have the authors provided feedback to the source databases about this? It would be extremely valuable so that future researchers didn't have the same problem and need to go through the same exercise.

6. In section 3.1: It is striking that the KEGG-MetaCyc model identified ~1,200 genes that were absent from the HMR2-based model (i.e. ~400 more genes that were shared between the models!), despite both methods being based essentially on sequence similarity. The authors note that one reason for the discrepancy is the presence of drug/xenobiotic pathways in the KEGG-MetaCyc model, but can the authors describe/speculate on the reasons for the other differences?

7. Lines 349: "Gene Orthology IDs" should be "Gene Ontology IDs"

8. Line 350 onwards talks about the "subcellular localization of genes". All these instances should be changed to refer to "gene products".

9. It is laudable that the authors consulted 9 different databases to obtain and integrate cellular compartment information, as described in section 3.1. But it would be useful if the authors could state if such a wide survey was required (in retrospect), and it is important to describe how any discrepancies between these sources were resolved.

10. Section 3.5, particularly, is overly lengthy - more than 4 pages of text associated with one figure (Fig 6). This section could

be substantially shortened without losing any essential content in order to improve its readability/accessibility.

11. Many definite/indefinite articles (ie. 'the' or 'a') are missing throughout the manuscript. E.g. the last sentence of the Introduction needs a "the" inserting prior to "...molecular basis" of.... The manuscript should be checked and these added where necessary.

12. Fig 4: Please state the classification system used for the given pathways in this Figure? Was it KEGG?

13. Fig 5: It would be useful if the names of the three models were used as labels above each graph, rather than having to refer to the legend.

Response to Reviewer Comments

Reviewer #1

In the present manuscript, Cesur and coworkers present a new genome-scale metabolic network reconstruction of *Drosophila melanogaster*, to facilitate the study of metabolism in this organism, which may give new insights into important biological processes (including development and neurodegenerative diseases). This reconstruction, to my knowledge, is the most comprehensive manually curated genome-scale reconstruction for the metabolic network of fruit fly, which also includes cell compartmentalization. I can recommend this manuscript for publication in case the authors address the following issues.

The authors are grateful for the positive and insightful comments and suggestions. Their comments have been addressed as detailed below, and the manuscript has been amended accordingly.

1. It is not completely clear to me how the authors investigate (and then, fix the problem of) the metabolic leaks. Firstly, the metabolic leaks are presumably a result of the presence of mass-unbalanced reactions (or, the so-called metabolic inconsistencies; see: DOI: 10.1093/bioinformatics/btn425). Did the authors find such inconsistencies in the network? Secondly, the authors mention that "To overcome this issue, we performed multiple reaction deletions and identified the reactions causing metabolic leaks. The identified reactions were manually curated". To me, this procedure is rather ambiguous. I strongly recommend the authors provide details for this procedure.

We agree with the reviewer about the requirement for further clarification of this approach. We determined metabolic leaks by searching for the presence of metabolites that can be produced in the absence of the uptake of any substrates. After we constrained all the uptake reactions to zero flux, we first maximized the rate of the production of a metabolite to see if that metabolite has leakage issue. If its production from 'nothing' is nonzero, we applied double-reaction deletions with the maximization of the production of that metabolite to see the pairs of reactions whose inactivation stops the leakage. Such reactions were identified as the cause of metabolic leaks. The reaction pair whose deletion resolves this issue was manually checked in terms of its metabolites, and revised. These steps were explained in the updated manuscript in more detail, lines 190-205:

"We also investigated the presence of leaking energy metabolites in the draft model. These metabolites represent the compounds that can be produced even in the absence of any nutritional intake. For instance, leaking ATP metabolite can lead to spontaneous energy production without any nutrient uptake, and it can result in a higher growth rate. In the leak testing, we analyzed the metabolism of the charging energy metabolites listed by Fritzsche and colleagues (e.g., ATP, NADH, NADPH, FADH₂, GTP, and H⁺) (Fritzsche *et al.*, 2017). To this end, we first constrained all uptake reactions in the model to zero by allowing only their secretions. Then, a dissipation reaction was added for each energy metabolite if the corresponding reaction was not available in the model. Maximization of the rate of each dissipation reaction was defined as an objective function to evaluate whether the corresponding metabolite is leaking or not. A nonzero rate for the dissipation reaction indicated the leakage of that metabolite. To overcome this problem, we performed double reaction deletions by setting

their lower and upper boundaries to zero under the same nutritional condition, where all the uptake reactions were constrained to zero flux. The reaction pairs whose deletion prevented the production of a leaking metabolite in the absence of nutrients were carefully revised by removing incorrectly compartmentalized or redundant reactions.”

2. I strongly recommend the authors use the MEMOTE server to validate their model, (and possibly, report the qualification results as a supplementary file).

We would like to thank the reviewer for their suggestion. The MEMOTE test suit was used to evaluate the model performance in a comparative manner with the reconstructions in the BiGG Models database. This analysis was applied as explained in the Materials and Methods section, lines 226-228:

“The quality of the iDrosophila1 was evaluated via the metabolic model test suite, MEMOTE (version 0.13.0) using the Gurobi solver (Gurobi Optimization, LLC)”.

The total MEMOTE score size is comparable with the reconstructions stored in the BiGG Models database (Fig S2 in the revised manuscript). The manuscript was updated accordingly, lines 519-525 and a supplementary figure reporting the results (Fig S2):

“The final model, called iDrosophila1, includes 8,230 reactions (5,787 enzymatic and 2,443 non-enzymatic), 6,990 metabolites, and 2,388 genes. The compatibility of the model with recommended standards was assessed using the MEMOTE test suite. Norsigian et al. reported the MEMOTE scores of 108 reconstruction models in the BiGG Models database, most of which are prokaryotic models (Norsigian et al, 2020). The iDrosophila1 model has a comparable score with the other 108 models (Fig S2).”

The legend of the newly added figure:

“Figure S2. Plot depicting MEMOTE scores of iDrosophila1 and the 108 metabolic reconstructions stored in the BiGG database. The reference organism and MEMOTE scores are reported for each metabolic reconstruction.”

3. In line 110, it is mentioned that "If a human gene has over nine *Drosophila* orthologs, this gene association was ignored". Please elaborate on the reason behind this step.

We appreciate the reviewer’s suggestion. The reason behind this step was elaborated in the current version of the manuscript as follows, lines 119-123:

“When a human gene has over nine *Drosophila* orthologs, this gene association was ignored (Blais *et al.*, 2017) since the orthology association is not specific. However, the related reactions were kept in the model since the existence of *Drosophila* orthologs showed that the corresponding reaction is available in *Drosophila*.”

4. Please explain the importance of aspartate in fruit fly metabolism (line 210). Why this specific amino acid is studied in this work (but not the other amino acids)?

Aspartate is a non-essential amino acid whose increasing level was reported to have no considerable effect on the growth rate of *Drosophila* (Schönborn *et al.*, 2019). Therefore, this amino acid was considered in the study. In the updated manuscript, we also demonstrated the effect of ten essential amino acids on the growth rate, lines 594-613:

“Aspartate is a non-essential amino acid whose deficiency was reported to have no detrimental effect on the lifespan of *Drosophila* (Piper *et al.*, 2014). Schönborn and colleagues reported that the increasing level of the aspartate amino acid did not affect biomass production (Schönborn *et al.*, 2019). We also examined the impact of varying aspartate levels on fly growth in the expanded HD condition in the iDrosophila1, FlySilico, and Fruitfly1 models. The iDrosophila1 model revealed that biomass production was not affected by aspartate depletion thanks to its inherent aspartate biosynthesis system. Besides, supplementation with additional aspartate did not enhance the growth at a considerable level (Fig 5D). This is consistent with FlySilico (Fig 5E) and Fruitfly1 (Fig 5F) simulations (Fig 5E). Further, *Drosophila* features ten essential amino acids, which were reported to be commonly essential between mammals and insects, except for arginine (Croset *et al.*, 2016; Manière *et al.*, 2020). We affirmed their essentiality using the iDrosophila1 model. The elevating intake of each essential amino acid contributed to an increase in the growth rate (Fig S3). Altogether, we confirmed the growth profile of fly across the varying levels of cholesterol and amino acids using iDrosophila1.”

5. About the study of gene knockouts (lines 226-228) there are at least two important issues. At first, a cutoff of <1% for inferring the lethality of a gene makes sense in the case of microorganisms, but not necessarily in the case of complex multicellular organisms like *D. melanogaster*. Secondly, the network reconstruction here represents the "generic" metabolism of fruit fly, while essentiality is a cell-type specific concept. I recommend the authors to, at least, mention these issues as the potential pitfalls of their approach to studying gene essentiality.

We thank the reviewer for raising these points. Indeed, there is no standard threshold value to decide the essential genes for multicellular organisms, as the reviewer pointed out. However, we agree that a cut-off based on higher percentages would make more sense for multicellular organisms. For instance, Khodaei *et al.* determined the essential genes in their mouse models (iMM1865 and min-iMM1865) using 30% of the optimal growth rate as a threshold value (Khodaei *et al.*, 2020). Therefore, we repeated our simulations with a cut-off of 30% for the *Drosophila* models, and we obtained the same essential gene sets as the ones we obtained with the cut-off of 1%. In the revised manuscript, the cut-off value was updated to 30% based on the paper of Khodaei *et al.*, lines 265-271):

“To evaluate gene essentiality, different cut-off values were preferred in the previous studies related to prokaryotic and eukaryotic metabolic models (Pratapa, Balachandran and Raman, 2015; Khodaei *et al.*, 2020; Wang *et al.*, 2021). Khodaei *et al.* determined the essential genes in their mouse models using 30% of the optimal growth rate as a threshold value (Khodaei *et al.*, 2020). We used the same cut-off of 30% to predict essential genes. In this regard, when the deletion of a gene resulted in a significantly reduced growth rate (i.e., a smaller growth rate than the given cut-off), this gene was considered essential.”

We agree with the reviewer that the gene essentiality analysis across cell types is more reasonable because of the variations between cells. However, the OGEE database does not store such detailed information for *Drosophila melanogaster*. Still, as the reviewer pointed out, we added a remark in the revised manuscript about this point. The first paragraph in Section 3.4 was revised as follows, lines 615-621:

“Gene essentiality refers to the indispensability of genes for survival under specific growth conditions. This concept is especially suitable to analyze cell-type specific gene essentiality due

to cellular variations. On the other hand, it is often used to evaluate the predictive capabilities of the reconstructed generic metabolic models due to the lack of comprehensive cell-specific information for many eukaryotic organisms. We assessed the prediction performance of the generic *iDrosophila1* model by *in silico* single-gene knockouts, and tissue-specific analyses may result in higher number of the essential genes.”

6. To me, it is not completely clear how metabolite names and IDs are matched. Supplementary figure 1 also provides incomplete information on this issue. Please provide further details.

We thank the reviewer for the important suggestion targeting to improve the procedure comprehension. The approach was explained providing more detail in the revised version of the Supplementary Note:

“In the first approach, synonymous metabolites within the model were interrogated based on compound names (metabolite name and BioCyc name) and IDs (KEGG (Kanehisa and Goto, 2000), ChEBI (Degtyarenko *et al.*, 2008), PubChem (Kim *et al.*, 2021), and LIPID MAPS (Fahy *et al.*, 2007)). To this aim, the compound information was collected from available metabolic networks and several web servers including Metabolite Translation Service (Caspi *et al.*, 2016), MBROLE 2.0 (López-Ibáñez, Pazos and Chagoyen, 2016), MetaboAnalyst (Chong *et al.*, 2018), and Chemical Translation Service (Wohlgemuth *et al.*, 2010). Note that only three-star ChEBI IDs (parent IDs) that were manually curated by the ChEBI team were kept in the compound list by excluding less reliable ChEBI IDs. Using this list, we subsequently generated a paired compound name and ID (collectively called identifiers) dictionary by mapping the related IDs to the metabolites in the given model. Based on these identifiers, we matched the model metabolites with each other by three main criteria: (1) If two metabolites within the model have at least one common identifier and they do not have any common identifier with any other metabolites, they were accepted as synonyms. (2) When three or more metabolites were detected to have common identifiers, a filtering step was applied to decide the most possible synonymous metabolite pair. In this prioritization process, the metabolite pair with the maximum number of matched identifiers (i.e., the highest number of matched IDs) were assumed to be synonymous. (3) It should be noted that, in the case where more than two metabolites have common identifiers, there can be more than one pair of metabolites with the maximum number of identifiers. Such hits were further investigated to select the most possible synonymous metabolites based on the databases including BioCyc (Karp *et al.*, 2018), ModelSEED (Seaver *et al.*, 2021), ChemSpider (Pence and Williams, 2010), and MetaNetX (Moretti *et al.*, 2021) in addition to the literature. Briefly, our approach focuses on the identification of the most potential synonyms considering the number of common identifiers and extensive knowledge-based manual curations to increase the accuracy by avoiding wrong matches derived from obsolete or incorrectly assigned IDs (Fig S1). For each metabolic match, one of the duplicated metabolites was removed from the model after the assembly of their stoichiometric coefficients in the model. This curation step was performed for the template human model, HMR2-based draft *Drosophila* model, and KEGG-MetaCyc-specific network.

The only difference between the first and second approaches is that they are dedicated to within- and between-model compound matches, respectively. In the second approach, we uncovered synonymous metabolites found in different metabolic models to successfully merge these networks. This step was employed before combining the HMR2-based and KEGG-MetaCyc-

specific metabolic networks to prevent any metabolic redundancy in the merged model. Similar to the previous approach, a compound name-ID pair dictionary was generated for each model. Then, the metabolites in both models were mutually matched based on the specified identifiers. The same assumption was used to prioritize the multi-matched metabolites. We selected the most reliable synonyms with the maximum number of common identifiers (Fig S1). After the manual confirmation of the potential synonyms, the names of synonymous metabolites in the KEGG-MetaCyc-specific network were replaced with the metabolite names in the HMR2-based draft *Drosophila* model. Thus, duplicated metabolites in the merged model could be easily distinguished and removed in the following step.”

Similarly, the legend of Fig S1 was updated accordingly:

“Figure S1. Identification and filtering of the synonymous metabolites by generating dictionaries including compound names (metabolite name and BioCyc name) and IDs (KEGG, ChEBI, PubChem, and LIPID MAPS) for the synonymous metabolite pairs. To determine the synonymous metabolites within a model, a single dictionary is generated using several sources (Metabolite Translation Service, MBROLE 2.0, MetaboAnalyst, Chemical Translation Service, and available metabolic networks). Based on the matched identifiers in the dictionary, single and multiple hits are determined. If a model compound has only one synonym as in the first illustrative example (see the metabolites M_1 and M_2), they are accepted as synonymous metabolites. For the multiple matches, the number of common identifiers is used to select the synonymous metabolites. The metabolites M_3 and M_4 share the maximum number of identifiers, so they are accepted to be synonyms in the second case. On the other hand, when there is more than one metabolite pairs sharing the same number of the maximum common identifiers, they should be carefully examined to decide the correct compound pair. In the last illustrative example, both M_1 and M_2 have three common identifiers with M_3 . Therefore, the most possible synonym of the M_3 should be selected and confirmed through the literature and available databases. The same workflow steps are applied to identify the synonymous metabolites between different models. The only difference is to compare multiple dictionaries due to the creation of a different dictionary for each model”.

Reviewer #2

This paper presents a new, thorough and comprehensive metabolic network model for *Drosophila melanogaster*. The first two sections of the Results describe how the new model was derived through computational and manual curation steps. The following sections validate and apply the model with respect to: i) the effect of increasing cholesterol/aspartate levels; ii) gene essentiality; and iii) metabolic phenotypes associated with Parkinson's disease. Overall, this is an advance on previous published models and is a welcome addition to the resources available to researchers of *Drosophila* metabolism. I judge the paper to be suitable for publication after several points have been addressed, most notably a clearer account of which parts of the model are based on computational (orthology-based) predictions versus direct evidence from *Drosophila*, and a clearer statement about how the computational model can be accessed/used. (I will note that I am not an expert in constructing/validating metabolic models so I cannot critique those technical aspects.)

The authors are grateful for the reviewer's positive comments on the manuscript. The constructive points raised from the reviewer have been addressed and the manuscript has been revised accordingly.

Major points to address:

1. The current title suggests the metabolic model is specific/restricted to the neural system, which is not the case and thereby undersells the scope and applicability of the model. Suggest the last 6 words of the title are either omitted or modified.

We appreciate the reviewer's suggestion. The title is now revised as follows:

"A metabolic model of *Drosophila melanogaster* and its application to the analysis of Parkinson's disease transcriptome"

2. As far as I can tell, the only way for researchers to access/use the iDrosophila1 model is via the GitHub repository that is mentioned discretely at the end of Methods section 2.1. This availability statement needs to be much more prominent - ie. mentioned in the Abstract and/or Results section. It would also be useful to include the following sentence from the Github site within the manuscript: "The model is shared in MATLAB .mat, .xml (SBML), and .json formats, which are compatible with COBRA Toolbox model structure".

In the updated manuscript, the link to the related GitHub website was also shared in the Abstract section. In addition, we provided information about the file formats in lines 224-226 (please also see the Data Availability section):

"The model can be accessible in MATLAB (MAT), XML (SBML), and JSON formats, which are compatible with the COBRA Toolbox model structure."

3. The primary *Drosophila* model is based on a human metabolic model called 'HMR2'. Assuming there are multiple human models available, can the authors say why they chose to use this one? E.g. is it the most recent, or the most comprehensive, or perceived as the 'best' for some reason? A few more details about HMR2 would also be appreciated - is the model based entirely

on empirical data from human cells, or are aspects of it computed from studies on other organisms?

We included the citation of HMR2 in the manuscript (Introduction section and Section 3.1). The previous knowledge about the template model was also expanded in the updated version, lines 345-359:

"The template model, an improved version of the HMR2, incorporates several improvements based on the experimental results and the available sources (e.g., previous GMN models) for accurate phenotype predictions and appropriate contextualization (Zirngibl, 2021). The HMR2 is the updated version of the human metabolic reaction (HMR) database (Agren et al, 2012), derived from the Edinburgh human metabolic network (Hao et al, 2010), Recon1 (Duarte et al, 2007), and external databases (Mardinoglu et al, 2014). The modifications in the HMR2-derived template human model mainly include the addition of mitochondrial intramembrane space, the removal of atomically imbalanced reactions, the curation of GPR associations, and the revision of reactions from the beta-oxidation pathway. The addition of the mitochondrial intramembrane space is particularly promising for predictions on respiratory ATP synthesis (Zirngibl, 2021). We further curated the GPR rules of the template model based on the information about protein complexes in the iHsa model (Blais et al, 2017). Thus, over 300 GPR associations were curated. In the reconstruction process of the draft fly model, we used metabolic information in the curated human model."

4. In several instances, the main text appears to suggest that the KEGG-MetaCyc model is not an orthology-based model, but is bringing in *Drosophila*-specific, perhaps empirical, information. E.g. in the abstract: "The gene coverage and metabolic information of the orthology-based draft model were expanded via *Drosophila*-specific KEGG and MetaCyc databases"; or from the introduction: "we first developed a draft model through the orthology-based mapping of *Drosophila* genes. Then, the draft *Drosophila* model was expanded based on metabolic information in KEGG and MetaCyc databases"; or from the second paragraph of section 3.1: "Using metabolic information in KEGG and MetaCyc databases, a KEGG-MetaCyc metabolic network was also generated for *D. melanogaster*". But the Methods section says that the authors "generated another metabolic network for *D. melanogaster* using RAVEN toolbox [21] by considering protein homology against KEGG [22] and MetaCyc [23] databases". Thus, the KEGG-MetaCyc model was generated by a similar method to the "orthology-based model", in that gene-protein-reaction rules from a curated set were transferred to the *Drosophila* genome based on sequence similarity in both cases. I recommend that: (i) the two models are referred to as the 'HMR2-based model' and the 'KEGG-MetaCyc model', to avoid distinguished the former as the only 'orthology-based model'; and (ii) the authors provide a clearer description of which reactions/pathways in the final iDrosophila1 model are based purely on orthology/sequence similarity, versus any that they manually curated from the *Drosophila* literature and thus have direct experimental evidence.

We thank the reviewer for raising this point. For the sake of clarity, we used the suggested nomenclature (i.e., 'HMR2-based model' and 'KEGG-MetaCyc model') for the reconstructed draft models.

The manually revised reactions were explained along with other curations in lines 431-433:

"In the next step, the gap-filled model was curated in terms of cholesterol and apocytochrome-C metabolism, GPR rules, metabolic leaks as well as the common curation steps described in Supplementary Note."

The curation of cholesterol metabolism by removal of the non-conserved human reactions from the *Drosophila* model allowed us to mimic the cholesterol auxotrophic phenotype of flies. In addition, the amendment of the apocytochrome-C metabolism by adding the apocytochrome-C metabolite and the related reactions affected the growth rate due to the presence of cytochrome-C in the 'cofactor and vitamin' composition of the biomass equation.

5. In section 3.5 (and Figure 6B), it's unclear whether the genes/pathways identified as 'enriched' are up- or down-regulated in the in the parkin or pink1 mutants. Please clarify.

Enriched terms include both upregulated and downregulated genes/pathways. The tools provided all enriched terms regardless of the regulation types. The manuscript has been amended accordingly, lines 748-750:

"The genes involved in these reactions were characterized by detecting enriched metabolic pathways (Fig 6B; also, see Table S8A-S8D for all enriched terms), which are either up- or down-regulated in the mutants."

6. The last paragraph of section 3.5 describes analysis of "enriched diseases associated with [*Drosophila*] pink1 and parkin mutations" (Fig 6C). It's unclear what it means for a disease to be associated with mutations in these fly genes, or how this association was determined. I couldn't find an explanation of this in the Methods. Please clarify.

We appreciate the reviewer's suggestion. In this section, we first determined the reactions regulated by *pink1* and *parkin* mutations. Then, the genes involved in these reactions were characterized according to the diseases significantly enriched with these genes. FlyEnrichr database was used for gene-disease association. This database is an extension of the Enrichr, which includes gene set libraries associated with various diseases from different sources. Using Fisher exact test, input gene lists are compared with these gene set libraries to reveal enriched diseases (Xie *et al.*, 2021).

In this regard, we provided our gene lists (the genes in the regulated reactions by the mutations) as inputs to identify over-represented diseases with default parameters. For the sake of the clarity, the first sentences of the last paragraph of Section 3.5 was updated, lines 853-859).

"Lastly, we determined diseases significantly enriched with the genes involved in the regulated reactions by *pink1* and *parkin* mutations. To do so, we compared the gene list with the gene set library in FlyEnrichr derived from 'Human Disease data from FlyBase (2017)', which provides disease-gene association information. Neurodegenerative disorders, muscular diseases, and mitochondrial diseases were found to be over-represented with the genes we identified, further confirming the accuracy of transcriptome-guided iDrosophila1 predictions (Fig 6C)."

Minor points to address:

1. I think this sentence in the introduction is wrong: "The highly conserved disease pathways of human have been extensively analyzed based on the *Drosophila* genes, which share 75-77% homology with disease-associated human genes [3, 4]." The relevant sentences from the given references are "about 75% of the genes responsible for human diseases have homologs in flies"

and "nearly 75% of human disease-causing genes are believed to have a functional homolog in the fly", which is a different statistic.

We thank the reviewer for raising this point. As the Reviewer highlighted, the references in the manuscript reported the existence of 75% homology with disease-associated human genes. On the other hand, Reiter and colleagues (2001) determined that 77% of OMIM human disease gene entries were significantly matched with *Drosophila* genes using systematic BLAST analysis (E-value $\leq 10^{-10}$). This article was also cited in the current manuscript by defining the homology level between 75% and 77%.

2. Please define what the 'i' in 'iDrosophila1' represents.

The first letter "i" represents "*in silico*". This abbreviation is commonly used in the denomination of genome-scale metabolic network models such as iMM1865, iYO844, and iCEL1273.

3. It's mentioned in section 3.1 and in the supplemental note that gene duplicates (paralogs) encoding the same enzyme were reviewed so that only one gene was present in the gene-protein-reaction and the final model. The note should mention how the 'winning gene' was chosen, and it would be useful to have a supplemental table listing these cases. Such paralogs may have important tissue or context-specific roles and are not necessarily 'redundant'.

We realized that the "duplicated" term is confusing in that section. Therefore, it was replaced by "redundant" in Section 3.1 and Supplementary Notes. The gene-centric curations include the revision of redundant genes and their repeated associations in each GPR rule. This redundancy emerged because of the ID conversions in the draft models. For instance, two different human genes in the same GPR rule can have the same *Drosophila* ortholog. When these genes are replaced by their *Drosophila* ortholog in the HMR2-based model reconstruction process, a redundancy occurs in the related GPR rule due to the repetition of the orthologous gene. Such redundancy must be removed in the curation step, as explained under the subtitle 'Gene-centric curations' in the Supplementary Note.

4. It would be helpful to mention at the end of the first paragraph of section 3.1 that the HMR2-based model includes intracellular compartments, rather than only introducing this point when describing the KEGG-MetaCyc model later.

We would like to thank the reviewer for their suggestion. The first paragraph of Section 3.1 was updated as follows:

"Importantly, the HMR2-based template human model contains eight intracellular compartments (cytosol, nucleus, Golgi apparatus, endoplasmic reticulum, mitochondria, mitochondrial intermembrane space, lysosome, and peroxisome) along with their Gene Ontology IDs, and this information was also transferred to the draft *Drosophila* model. The reconstructed draft model includes 6,873 reactions, 4,856 metabolites, and 1,321 genes."

5. Section 3.1 and the supplemental note describe the need for and utility of cataloguing the different names/IDs for the same metabolite in different source databases. E.g. the supplementary note says "potential synonyms were subsequently revised via an extensive manual curation step to avoid wrong matches due to obsolete or incorrectly assigned IDs." Have the authors provided feedback to the source databases about this? It would be extremely

valuable so that future researchers didn't have the same problem and need to go through the same exercise.

We thank the reviewer for this constructive remark. We contacted the databases, and we are thinking of sharing our dictionary with their curators to provide helpful feedback.

6. In section 3.1: It is striking that the KEGG-MetaCyc model identified ~1,200 genes that were absent from the HMR2-based model (i.e. ~400 more genes that were shared between the models!), despite both methods being based essentially on sequence similarity. The authors note that one reason for the discrepancy is the presence of drug/xenobiotic pathways in the KEGG-MetaCyc model, but can the authors describe/speculate on the reasons for the other differences?

We thank the reviewer for their comment. In the reconstruction of the draft *Drosophila* model from the reference human model, the orthologous genes were identified based on DIOPT scores. Only orthologous *Drosophila* genes with the highest DIOPT scores were kept in the model by excluding the remaining matched genes. This strict approach might have led to the elimination of some orthologous *Drosophila* genes.

Additionally, sequence similarity in our draft *Drosophila* model was derived from the HMR2-based reference model. Its first version (HMR) was developed by incorporating the existing metabolic information from the Recon1 model (Duarte *et al.*, 2007), Edinburgh Human Metabolic Network (Hao *et al.*, 2010), and several databases (Agren *et al.*, 2012). The HMR2 was subsequently reconstructed by improving lipid metabolism in the HMR (Mardinoglu *et al.*, 2014). The missing information in the HMR2 (and so the reference human model) might be another reason behind the missing *Drosophila* genes in the orthology-based gene mapping step. Therefore, the use of information in up-to-date KEGG and MetaCyc databases is especially useful to increase the gene content in the model.

Another reason is that the *Drosophila*-specific metabolic information derived from the KEGG and MetaCyc databases has been revised by the curators independently from the HMR2-based reference human model. Thus, these networks can harbor different metabolic knowledges.

7. Lines 349: "Gene Orthology IDs" should be "Gene Ontology IDs"

We thank the reviewer for their careful reading of the manuscript. The wrong expression was corrected in the current manuscript.

8. Line 350 onwards talks about the "subcellular localization of genes". All these instances should be changed to refer to "gene products".

We thank the reviewer for this point. We carefully revised all these instances in the manuscript.

9. It is laudable that the authors consulted 9 different databases to obtain and integrate cellular compartment information, as described in section 3.1. But it would be useful if the authors could state if such a wide survey was required (in retrospect), and it is important to describe how any discrepancies between these sources were resolved.

We would like to thank the reviewer for their valuable comment. In the compartmentalization process, we preferred to use several sources to assign at least one subcellular localization for

each gene product. We wanted to follow a flexible approach since it is known that an enzyme can be localized in multiple compartments.

As described in the reaction-centric curation step (Supplementary Notes), the reactions shared by both KEGG-MetaCyc and template human model were identified iteratively by ignoring compartment information in the models due to the flexible compartmentalization of the KEGG-MetaCyc network. If there is any compartmental inconsistency between the duplicated reactions, we relied on the HMR2-based compartment information in the template model. In this way, the compartmental discrepancies were eliminated by removing the shared KEGG-MetaCyc reactions, which had different compartments from the template model.

10. Section 3.5, particularly, is overly lengthy - more than 4 pages of text associated with one figure (Fig 6). This section could be substantially shortened without losing any essential content in order to improve its readability/accessibility.

We agree with the reviewer, and we did our best to shorten Section 3.5 in the updated manuscript.

11. Many definite/indefinite articles (ie. 'the' or 'a') are missing throughout the manuscript. E.g. the last sentence of the Introduction needs a "the" inserting prior to "...molecular basis" of.... The manuscript should be checked and these added where necessary.

We thank the reviewer for their careful reading of the manuscript. We checked and revised the manuscript grammatically.

12. Fig 4: Please state the classification system used for the given pathways in this Figure? Was it KEGG?

The given pathway (subsystem) information was obtained from the HMR 2.0 database. This information was added to the legend of Fig 4 to clarify this point:

“Figure 4. Distribution of the pathway and compartment information in the iDrosophila1 model. (A) HMR2-based metabolic pathways in the model are ordered according to their frequency and the top ten pathways are represented. The pie chart indicates the percentage of compartments in the (B) template human model and (C) iDrosophila1.”

13. Fig 5: It would be useful if the names of the three models were used as labels above each graph, rather than having to refer to the legend.

We thank the reviewer for their suggestion. The figure was updated according to the reviewer's comment. The new version is reported below.

May 2, 2023

RE: Life Science Alliance Manuscript #LSA-2022-01695-TR

Prof. Tunahan Çakır
Gebze Technical University
Gebze Technical University, Department of Bioengineering
Kocaeli 41400
Turkey

Dear Dr. Çakır,

Thank you for submitting your revised manuscript entitled "A new metabolic model of *Drosophila melanogaster* and the integrative analysis of Parkinson's disease". We would be happy to publish your paper in Life Science Alliance pending final revisions necessary to meet our formatting guidelines.

- please add the author contributions and a conflict of interest statement to the main manuscript text
- please add a Running Title for your manuscript to our system
- please add the Twitter handle of your host institute/organization as well as your own or/and one of the authors in our system
- please add a conflict of interest statement to your main manuscript text
- please incorporate your conclusion section at the end of the manuscript text into your main manuscript
- please consult our manuscript preparation guidelines <https://www.life-science-alliance.org/manuscript-prep> and make sure your manuscript sections are in the correct order
- please add the panel C to your Figure 3 figure legend

A. FINAL FILES:

B. MANUSCRIPT ORGANIZATION AND FORMATTING:

Sincerely,

Reviewer #1 (Comments to the Authors (Required)):

Now, I endorse the publication of this paper.

Reviewer #2 (Comments to the Authors (Required)):

The authors have addressed the comments I made in my review of the original manuscript, and I now recommend publication.

May 12, 2023

RE: Life Science Alliance Manuscript #LSA-2022-01695-TRR

Prof. Tunahan Çakır
Gebze Technical University
Gebze Technical University, Department of Bioengineering
Kocaeli 41400
Turkey

Dear Dr. Çakır,

Thank you for submitting your Research Article entitled "A new metabolic model of *Drosophila melanogaster* and the integrative analysis of Parkinson's disease". It is a pleasure to let you know that your manuscript is now accepted for publication in Life Science Alliance. Congratulations on this interesting work.

DISTRIBUTION OF MATERIALS:

Again, congratulations on a very nice paper. I hope you found the review process to be constructive and are pleased with how the manuscript was handled editorially. We look forward to future exciting submissions from your lab.

Sincerely,
